# Historic transposon mobilisation waves create distinct pools of adaptive variants in a major crop pathogen

Tobias Baril ✉, Guido Puccetti & Daniel Croll ✉

Transposable elements (TEs) shape host-pathogen interactions and contribute to antimicrobial resistance, but how adaptive TEs arise in populations and how historical contingencies affect TE dynamics remains unclear. Fungal crop pathogens provide unique frameworks to address such questions due to spatially explicit sampling and well-characterized niches. We characterise TE evolutionary dynamics in 1953 publicly available genomes across the global distribution range of the major wheat pathogen *Zymoseptoria tritici*. We characterise genomic diversity, benchmark methods to infer TE insertion polymorphism, and assess TEs as a source of adaptive variation. Across ~3.2 million annotated TE loci, we find substantial variation in TE content within and across populations. TE activity surged during the pathogen's expansion from the Middle East, with distinct activity profiles in derived populations. TE-mediated adaptation emerged from waves of TE mobilization, with the highest TE activity observed over as little as 25 years. 45 TE loci within 1 kb of 49 host genes show local adaptation signatures, likely related to adaptation to anti-fungals and the plant host environment. This work highlights the power of vast genomic datasets to unravel intraspecies TE invasion histories and pinpoint factors driving recent adaptation and argues for deep population-level TE surveys to uncover molecular drivers of adaptive evolution.

Exposure to new environments can shift fitness optima and challenge naïve species to adapt. The ability to rapidly respond to novel stressors can provide a significant evolutionary advantage. Genomic variation arising through mutation can modify an organism's ability to survive and reproduce in changing environments, and much focus is given to the role of single-nucleotide polymorphisms (SNPs) in the emergence of adaptive variation, whilst transposable elements (TEs), DNA sequences capable of moving from one genomic location to another[1], are often ignored[2]. However, there is growing evidence highlighting the importance of TE-derived genomic variation underpinning traits such as xenobiotic resistance[3–5], stress responsiveness[6] and environmental adaptation[7]. Despite this, our understanding of the genome-wide role of TEs and the extent to which TEs are a general source of adaptive genomic variation remains largely unexplored.

TEs can represent significant portions of the genomes of eukaryotes, comprising over two-thirds of the human genome[8] and over 90% of the maize genome[9]. However, other lineages have been more successful in suppressing deleterious TE activity, with TEs contributing only 0.13% of total genome size in *Trichoplax* (currently the lowest content among Animalia)[10] and rarely over 5% of total genome size in yeasts[11,12]. Autonomous TEs encode domains for their transposition activity, rendering them candidate sources of genomic variation for rapid host adaptation by moving protein-coding and regulatory units around host genomes. TEs are also known to be activated under

Laboratory of Evolutionary Genetics, Institute of Biology, University of Neuchâtel, Neuchâtel, Switzerland. ✉e-mail: tobias.baril@unine.ch;
daniel.croll@unine.ch

stress[13–15], coupling exposure to novel environments with transposition activity to give rise to adaptive variation.

The effects of transposition events vary depending on TE insertion sites within host genomes. Through their duplication and mobilisation, TEs generate diverse mutations, including the generation and modification of gene regulatory networks, chromosomal rearrangements, exon shuffling, and donation of coding sequence through domestication and co-option[16–21]. TEs have facilitated rapid adaptive evolution through single transposition events across the tree of life. For example, TE insertions have led to the evolution of V(D)J recombination for adaptive immunity in jawed vertebrates[22], the transition from a commensal to pathogenic lifestyle in *Escherichia coli*[23], cold-stress adaptation in blood oranges[24], xenobiotic resistance in insects[3,4,25,26] and fungi[5,27,28], industrial melanism in the peppered moth[29], and effector genes essential for host infection in oomycetes and fungi[15,27]. TE activity is also likely a significant source of genomic variation at the population level, with intraspecies TE activity associated with adaptation to cold[7] and urban environments[30], as well as modifying inter-individual immune responses[31]. Beneficial TE insertions are expected to experience strong positive selection and rapid fixation in populations, whilst the majority of TE insertions are expected to range from nearly neutral to deleterious. TEs exerting highly deleterious effects are efficiently removed via purifying selection[32], whilst the efficiency of selection is reduced against TEs with less deleterious effects, such as non-autonomous and fragmented TE insertions[33,34].

Bursts of TE activity in response to environmental change are hypothesised to be important drivers of micro- and macro-evolutionary events, whilst generating and maintaining genomic diversity[35,36]. Recent work highlights the power of TEs to invade and subsequently propagate in host genomes to generate significant genomic diversity through repeated waves of TE activity over timescales as short as 50 years[36,37]. Crucially, evidence mechanistically linking such TE waves to adaptive evolution is currently lacking, and previous studies have been unable to systematically assess this on a global scale due to challenges associated with incomplete sampling and analytical limitations. Nonetheless, it is hypothesised that new alleles generated by waves of TE activity may become adaptive in response to environmental stress. Identifying adaptive TEs remains challenging, in part due to a lack of available large-scale population datasets to systematically characterise TE-host evolutionary dynamics with enough statistical power, combined with a lack of functional studies to mechanistically link candidates to adaptive phenotypes. Our understanding of TE-host dynamics has also been limited by a focus on a few model systems. However, genome dynamics following TE invasions vary significantly among eukaryotic lineages. For example, transition-transversion bias was considered universal across metazoans, but counterexamples exist[38]. Further, population-level studies often identify TEs with adaptive signatures but fail to link these signatures to the underlying historic context of TE-host dynamics. This leads to questions concerning the mode and tempo of TE-mediated adaptation. Whilst TEs are regarded as sources of variation for rapid adaptation, it remains challenging to date TE activity and characterise the timescales during which TEs provide an adaptive benefit.

Here, we focus on one of the best characterised fungal species in terms of TE content in the genome. The ascomycete *Zymoseptoria tritici* is a major globally distributed fungal pathogen of wheat, causing significant yield losses[39]. The centre of origin is located in the Middle East, where its emergence accompanied the domestication of wheat[40]. *Z. tritici* has shown rapid adaptive responses across all major wheat-producing areas to strong selective pressures, including host resistance and fungicide use[41]. Extensive standing genetic variation is observed within and among distinct global populations of *Z. tritici*[42–46], which provides the capacity to respond rapidly to novel stressors[6].

Specific examples of TE-mediated adaptation exist, including multi-drug resistance[5], epigenetic regulation of key genes involved in host plant infection[15], and melanin production as a response to stressful environments[6]. In parallel to the global expansion of the pathogen, TE content of the genome has likely expanded as a consequence of less efficient genomic defences[42,45]. The well-documented co-evolutionary history between *Z. tritici* and domesticated wheat presents a unique opportunity to explore TE-host evolutionary dynamics within defined timescales.

Here, we aim to determine the tempo and adaptive roles of intraspecies TE invasions using one of the largest intraspecies genome sequencing datasets for eukaryotes. We manually curate 331 distinct TE families and subsequently identified TE insertion and absence polymorphisms in 1953 publicly available genomes sampled across the global range of *Z. tritici*[42,46]. Using this extensive global panel, we characterise population-specific waves of TE activity throughout the pathogen's evolutionary history. We assess TE locus frequencies among populations and identify candidate TEs with signatures of local adaptation, integrating data on fungicide resistance and bioclimatic variables. Overall, we find repeated instances of TEs exhibiting signatures of positive selection in a population-specific manner and associated with distinct TE activation waves in the species history.

## Results

### Global reference genome panel-informed TE discovery and benchmarking

To maximise capture of intraspecies TE diversity in our analysis, we used a global panel of reference genomes for TE curation. Earl Grey[47] was used to de novo annotate the non-redundant genome assembly comprised of 19 reference-quality genomes representing the global diversity of *Z. tritici*, and four closely related sister species (*Z. pseudotritici, Z. passerinii, Z. ardabiliae*, and *Z. brevis*). We manually curated de novo-generated TE consensi, after which the final TE library contains 331 TE consensus sequences composed of 92 DNA transposons, 22 long-interspersed nuclear elements (LINEs), 65 long terminal-repeat retrotransposons (LTRs), 31 miniature inverted terminal-repeat elements (MITEs, non-autonomous DNA transposons), 11 rolling circles (also known as *helitrons*), 1 short interspersed nuclear element (SINE), 1 terminal-repeat retrotransposon in miniature (TRIM), and 105 unclassified elements[48]. The high number of unclassified elements reflects limited knowledge of TE diversity across Fungi, hindering confident classification of sequences lacking known diagnostic features or homology to TEs from distant taxa. Since host genes and segmental duplications were removed during manual curation, many unclassified sequences in the final library may represent true TEs. As fungal TE landscapes become better characterised, these elements are likely to be classified with greater confidence. Reference genome annotations reveal a TE content between 17.51% (IR01_26b from Iran) and 24.86% (Aus01 from Australia) of total genome size (Supplementary Fig. S1, Supplementary Data 4), GFF annotations are available at https://doi.org/10.5281/zenodo.8390461.

Detecting TE insertions in short-read whole genome sequencing data is inherently challenging due to the difficulty of assembling repetitive DNA sequences from short reads. To tackle this problem, we performed extensive benchmarking based on 11 methodologies implemented in the McClintock2 meta-pipeline[49]. Performance was highly variable, consistent with previous findings[49]. Across all benchmarks, the highest-performing methodologies were TEFLoN[50] and PoPoolationTE[51] for reference and non-reference TE insertions, and TEMP2[52] for non-reference insertions. We adopted an approach that combines the best-performing reference and non-reference TE detection tools (see "Methods"; Supplementary Data 1). We maximised TE detection for each isolate whilst minimising false positive rates (see Supplementary Data 1 for an in-depth discussion).

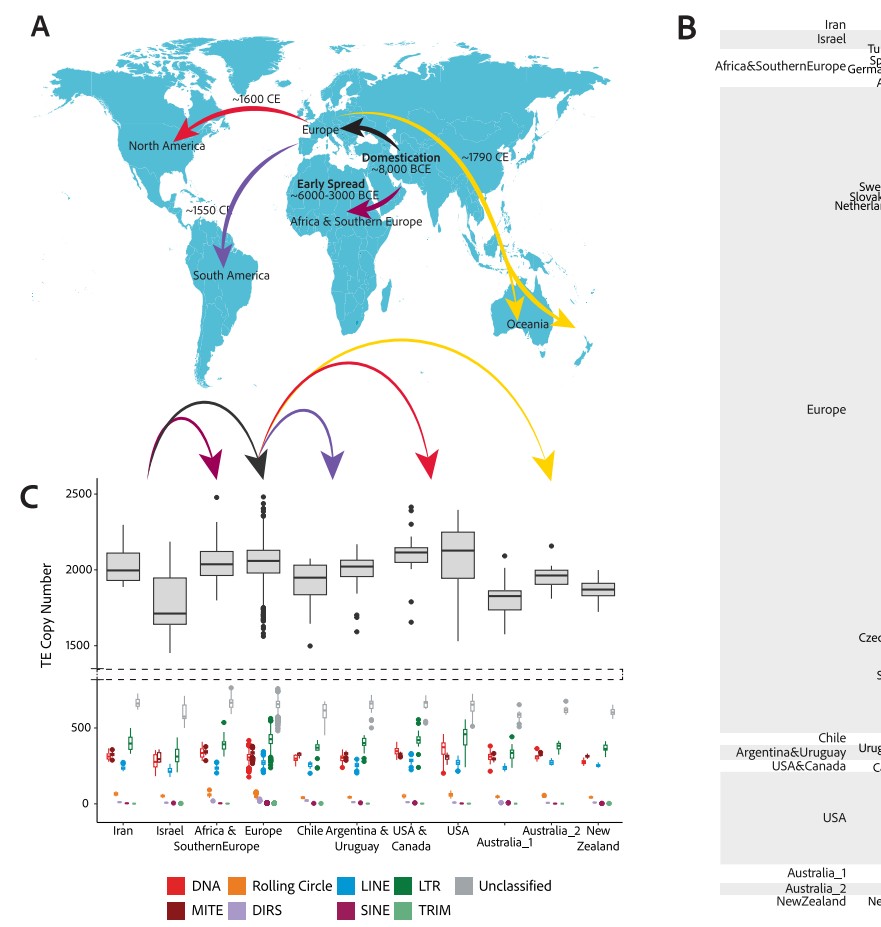

**Fig. 1 | Global dynamics of TE variation among populations of *Z. tritici*. A** A schematic illustrating the global spread of wheat across the globe. **B** TE copy number for each isolate (*n* = 1569) split by high-level TE classifications, as indicated in the key. Isolates are grouped by country of sampling and genetic cluster membership, ordered from most ancient to most recent using a colonisation timeline based on global wheat domestication. **C** Population-wide mean TE copy numbers. Top boxes represent total TE copy numbers. Bottom boxes show TE copy numbers split by high-level TE classifications, as indicated in the key. Box plot limits are set by 25th and 75th percentiles, with whiskers extending to the minimum and maximum observed values within 1.5 times the interquartile range (IQR) from their respective quartiles. Outliers are shown for observations more than 1.5 times IQR, indicated by points outside whiskers. Central lines of box plots indicate median values for each population. Population samples sizes: Iran *n* = 16; Israel *n* = 34; Africa & Southern Europe *n* = 39; Europe *n* = 1190; Chile *n* = 14; Argentina & Uruguay *n* = 28; USA & Canada *n* = 17; USA *n* = 171; Australia_1 *n* = 28; Australia_2 *n* = 7; New Zealand *n* = 25. Source data are provided as a Source Data file.

## A highly dynamic transposable element landscape following global expansion

We analysed 1953 publicly available isolates of *Z. tritici* sampled across the globe. Isolates were assigned to 12 genetic clusters based on genome-wide polymorphism, consistent with previous analyses[42,46] (Fig. 1A). Isolates that could not be clearly assigned to a single cluster (*n* = 384) were not included in analyses requiring explicit population assignment, retaining a total of 1569 isolates. From most ancient to most recent in terms of colonisation history, populations are named based on the origin of member isolates: i.e., Iran, Israel, Africa & Southern Europe, Europe, USA & Canada, USA, Chile, Argentina & Uruguay, Australia_1, Australia_2, and New Zealand.

Among all isolates, 3,201,910 TE loci were detected (μ = 2041; min = 1451, isolate ST92ISR_Ar_16h from Israel; max = 2481, isolate 08STCZ016 from Europe; Supplementary Data 2). There are significant differences in TE abundance among populations (One-way analysis of variance: $F_{10,1558}$ = 28.25, *p* < 0.01), illustrating the dynamic nature of TE landscapes among populations despite the separation by only short evolutionary timescales, with the Iranian and Israeli populations arising ~10,000 years ago, and the most recent Oceanian populations (Australia_1, Australia_2, and New Zealand) arising ~1790 AD. TE copy number has increased significantly as populations spread beyond the Middle East and colonised Africa, Europe, and the Americas, suggesting an increase in TE activity associated with colonisation of new environments (Fig. 1C; Tukey post-hoc comparisons are supplied in Supplementary Data 5; Supplementary Fig. S2). However, populations in Australia and New Zealand do not exhibit this pattern, showing a significant reduction in TE abundance compared to the Iran, Africa & Southern Europe, Europe, USA & Canada, USA, and Argentina & Uruguay clusters (Supplementary Data 5). The time of establishment of populations can be used to approximate the timing of TE activity, where novel TE loci arising in distinct populations are approximated to have arisen around the time of colonisation. Therefore, our observations are consistent with increased TE activity as *Z. tritici* spread out of the Middle East ~8000 BCE[42], into Africa (~6000–3000 BCE[42]), Europe, and the Americas (~1550–1600 CE[42]), followed by a reduction in TE activity, and TE degradation or removal as the pathogen colonised Oceania (~1790 CE) (Fig. 1).

In addition to significant variation among populations, inter-individual differences in TE abundance are also considerable in scale, with standard deviations in TE copy number ranging from 59 (New Zealand) to 203 (Israel) (Fig. 1C). Consistent with the reduced TE abundance observed in Oceanian populations, we also observe a decrease in TE abundance variation among Oceanian isolates. This

decrease could be attributed to the reduced level of genetic diversity observed in these populations as a result of the colonisation bottleneck. However, the TE decrease observed in Oceania may also be explained by a dampening of TE activity following counterselection against deleterious insertions.

In all populations, LTR retrotransposons are the most abundant TEs (global $\mu = 384$, min = 319 (Israel), max = 438 (USA)). Most populations share a similar profile in terms of TE abundance, characterised by high numbers of LTR retrotransposons (global $\mu = 384$), followed by MITEs (global $\mu = 321$), DNA TEs (global $\mu = 313$), LINEs (global $\mu = 254$), Rolling Circles (global $\mu = 50$), DIRS elements (global $\mu = 10$), SINEs (global $\mu = 2$), and TRIMs (global $\mu = 1$). However, in North America, DNA TEs have expanded to overtake MITEs as the second-most abundant TE classification (USA: DNA $\mu = 363$, MITE $\mu = 306$; USA & Canada: DNA $\mu = 347$, MITE $\mu = 328$; Australia_1: DNA $\mu = 307$, MITE $\mu = 296$) (Fig. 1, Supplementary Data 6).

Variation in observed TE numbers among the population could arise from genetic drift or relaxed purifying selection rather than TE activation. We investigated this possibility by analysing TE insertion frequency spectra for the most abundant TE families (Supplementary Figs. S3, S4). Contrary to expectations of relaxed purifying selection, we do not observe shifts in TE locus frequency spectra towards higher frequencies. This is consistent with TE insertion landscapes being shaped by ongoing TE activity (and purifying selection). In a second step, we considered the likelihood that all TE insertions detected in derived populations could have been present at very low frequency in the centre of origin populations and, hence, were not observable with the current sample size. We identified strong excesses of low-frequency insertions, inconsistent with undersampling in the centre of origin. This is supportive of ongoing transposition activity in derived populations. Finally, we considered genetic structure among isolates using either (largely neutral) SNP loci or TE presence/absence loci (Supplementary Fig. S5). We found strong incongruence between the genetic structures derived by the two sets of markers, which supports the idea that TE loci have differentiated in a non-neutral fashion. This is in accordance with TE activity rather than drift alone impacting TE variation among populations.

Beyond changes in TE activity during global expansion, we also observe large intra-population variation in TE copy number. High levels of intra-population variation are hallmarks of rapid evolution on very short timescales. Within the USA population, we find individuals falling into two groups characterised by TE abundances either below or above 2000 copies. These subpopulations highlight rapid increases in TE abundance within the USA population, where 96% of the isolates sampled in the 1990s contain <2000 TEs and 83% of the isolates sampled in the 2010s contain >2000 TEs. When comparing isolates sampled in the 1990s to those sampled in the 2010s, the mean TE abundance has increased from 1840 to 2227 TEs per isolate, driven mostly by LTR retrotransposon activity, consistent with previous findings (Oggenfuss et al.[45]), and DNA TE activity (LTR retrotransposons: $\mu_{1990's} = 372$, $\mu_{2010's} = 485$; DNA TEs: $\mu_{1990's} = 315$, $\mu_{2010's} = 398$) (Fig. 1B).

Given that TE activity is overall detrimental to host genome integrity, most TE families are expected to be repressed and maintained at low copy numbers. Consistent with this, 261 (79%) TE families in *Z. tritici* have a maximum population mean copy number <10. Conversely, we also observe large increases in the abundance of specific TE families as *Z. tritici* expanded from the centre of origin in the Middle East. 18 TE families show mean TE copy number variation of >10 among populations (Fig. 2B). The spread of *Z. tritici* into North America ~1550 AD has led to increases in TE activity, where LTR retrotransposons and rolling circle TEs increased in copy number to their highest levels in North American populations. More recently, DNA TEs have expanded in the Oceanian populations, representing the last

colonised continent. DIRS elements are found at very low numbers across all populations except for the Chilean population, where these have expanded 3-fold in copy number (Fig. 2B). In contrast, LINEs show a stepwise increase tracking the colonisation timeline of populations, suggesting low levels of ongoing activity rather than TE burst dynamics as observed for other TEs.

Beyond population-specific differences in total TE abundance, we observe substantial levels of population-specific TE activity. There are significant differences among genomes in their affinity for highly abundant TE families (i.e., defined as having ≥20 copies in at least 1 isolate; Pearson's $\chi^2_{118889} = 132,250$; $p < 0.01$; Fig. 2A). Clustering genomes based on their affinity for highly abundant TE families largely recapitulates the species' population history, which is indicative of population-specific TE-host dynamics as an important source of genomic differentiation (Fig. 2A). The most abundant TE family in the centre of origin populations is a LINE (*family_1507*; Iran $\mu = 64$; Israel $\mu = 61$). This family largely ceased activity during the global expansion of *Z. tritici*, persisting in populations at a mean abundance of between 61 (Israel and USA) and 66 (Africa & Southern Europe). In contrast, the *Copia* LTR retrotransposon, *family_71*, shows significant activity in the USA, Europe, Australia_1, and Australia_2 populations (USA $\mu = 81$, Australia_1 $\mu = 65$, Europe $\mu = 61$, Australia_2 $\mu = 67$) (Figs. 2B, 3B). This *Copia* LTR retrotransposon is the most active TE in *Z. tritici* but reached peak activity only after persisting at low copy numbers in early colonising populations (Iran $\mu = 17$; Israel $\mu = 10$, Africa & Southern Europe $\mu = 7$; Figs. 2B, 3B). The pattern observed for this *Copia* element is consistent with the TE burst model, whereby the exposure to abiotic stress in a novel environment during range expansion results in massive bursts of specific TE families[35]. Such large increases in TE copy number have the potential to severely impact host genome integrity, as demonstrated for the *Styx* element (*family_11*)[20], even whilst proliferating to a lesser extent than *family_71*. *Styx* is absent in the Iranian population and found at its highest mean copy number in Australia, expanding throughout the global expansion (Supplementary Data 7).

## TE contributions to a highly dynamic genome

The genome of *Z. tritici* is shaped by ongoing TE activity contributing to high levels of genomic diversity. To quantify the exact contributions to genetic variability, we assessed polymorphism created by the presence or absence of TEs across loci. 1477 insertion sites are found in at least one genome in every population, constituting a pool of persistent TE variants, whilst 214,312 sites are unique to genomes from a single population (Fig. 3A). Population-specific waves of TE activity produced 28,740 population-specific polymorphic TE loci arising in populations established in the last ~500 years with the USA contributing 21,120 loci alone (Supplementary Data 7).

The majority of TE loci show high TE insertion frequencies within populations (defined as the TE being at frequencies ranging 10–95%, also termed common insertions). The mean number of loci at such high TE frequencies is 1025 (49.85% of total TE loci; Europe) to 1370 loci (65.40% of total TE loci; USA). TE mean frequencies at the loci range between 54.64% (Iran) and 68.62% (Australia_1) (Supplementary Data 8). The number of loci carrying rare TEs per isolate is much lower and ranges between 100 (4.95% of total TE loci; Iran) and 407 (19.81% of total TE loci; Europe). Loci with fixed TEs ranged between 232 (13.15% of total TE loci; Israel) and 893 (45.63% of total TE loci; Australia_2). Large variation around the mean TE frequencies may underpin evolutionary potential for the species. TE activity has occurred over short evolutionary timescales, with most of the TE activation waves occurring in the last ~500 years. The recency of TE activation waves raises the question as to why such activity is tolerated by the genome and whether such activity could have produced adaptive benefits.

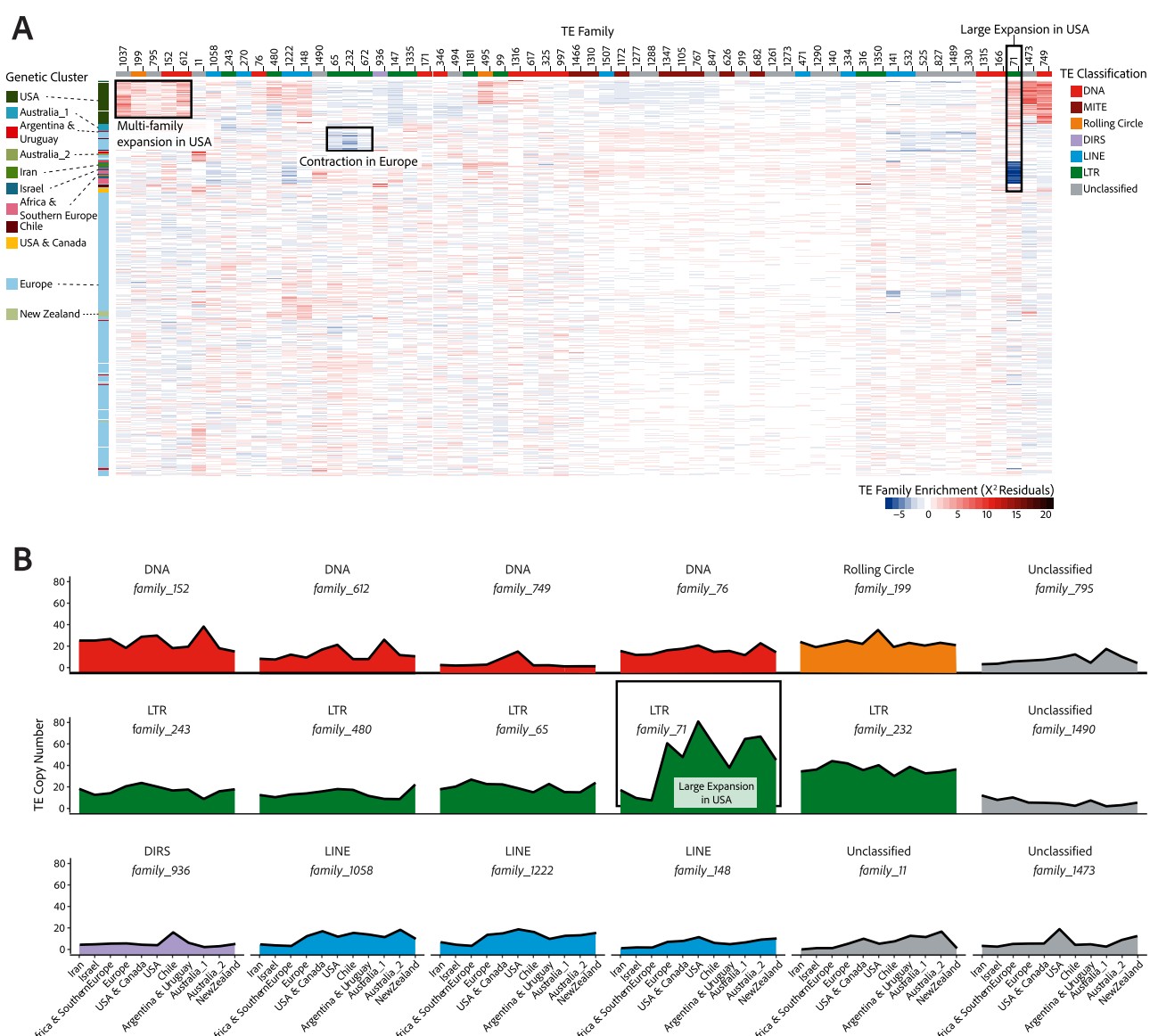

**Fig. 2 | TE activity gains and losses during global pathogen expansion. A** Affinity of individual isolates for the most abundant TE families (≥20 copies in at least one isolate) using Pearson's $\chi^2$ residuals. Negative affinities (i.e., repulsions) are illustrated with blue shades, whilst positive affinities (i.e., attractions) are illustrated with red shades, as indicated in the key. TE family classifications are illustrated on the top edge, along with TE family numbers corresponding to the species TE library[48]. Isolate population membership is illustrated on the left edge. Boxes highlight patterns of TE expansion or contraction as labelled. **B** TE burst profiles showing mean population-wide TE abundance for families with an abundance variation ≥10 among populations. Populations are ordered from most ancient to most recent based on the global spread of wheat. *Family_71*, which shows a large expansion in the USA population, is highlighted by a box in (**A**) and (**B**). Source data are provided as a Source Data file.

## Candidate adaptive TEs facilitating the global pathogen spread

TEs rising to high frequencies in specific populations are prime candidates underpinning recent adaptation, driven by positive selection in extant populations due to a positive effect on host fitness[53]. We identified the strongest candidate TEs with signatures of local adaptation by detecting TE loci with outlier differentiation among populations using BayPass[54]. BayPass is designed to identify genetic markers subject to selection using models that explicitly account for covariance structure among population allele frequencies that arises from the shared histories of the populations, whilst making no assumptions about the underlying demography.

We retained 45 consistently identified TE loci for further inspection (Fig. 4). Nine loci were discarded as the genomic context could not be confidently assessed. The 36 remaining putatively adaptive TE loci were proximal to 49 host genes within a 1000 bp window. We

identified conserved functions for 20 genes[55,56] (Fig. 4; Supplementary Data 9, 10). The most abundant GO terms included Zinc ion binding (GO:0008270; $n = 5$); ATP binding (GO:0005524; $n = 4$); Mitochondrion (GO:0005739; $n = 3$); Membrane (GO:0016020; $n = 3$) and ATP hydrolysis activity (GO:0016997; $n = 3$). Likely adaptive TE loci are found in the promoter and 5' UTR regions of 30 distinct host genes, where insertions have the potential to modify host gene expression (Fig. 4; Supplementary Data 10). Six host genes contain adaptive TEs in their coding sequence, whilst four genes contain candidate TEs in their 3' UTR and 12 genes carry TEs in their downstream region (Supplementary Data 10).

To assess the adaptive role of TEs in the pathogen's historic expansion, we assigned candidate adaptive TE loci to individual historic activation waves. Furthermore, we integrated know-how from a large antifungal resistance mapping study on *Z. tritici*, as

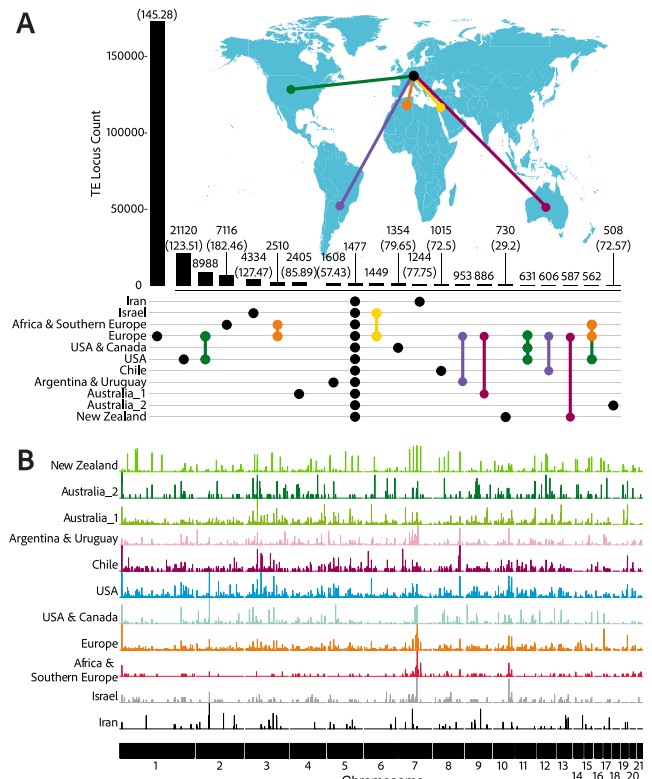

**Fig. 3 | TE activity associated with population differentiation across the globe.**
**A** TE loci identified in each population of *Z. tritici*, grouped by populations containing each TE insertion. Bars are in descending order by locus count. Dots indicate populations where at least one isolate contains the TE locus. Black lines connect populations that share TE loci under their respective quantification bars. Coloured connections indicate shared loci among populations indicated on the map. Numbers in brackets show TE loci normalised by sample size for population-specific loci. **B** Expansion of the highly active LTR, *family_71* and occupancy across genomic loci. Peaks indicate TE insertion frequency across chromosomes. Populations are ordered following global pathogen spread. Source data are provided as a Source Data file.

well as climatic data. Two candidate adaptive TEs are found within 1000 bp of 6 SNPs associated with fungicide resistance (Fig. 4). The LTR retrotransposon (*family_1437*) on chromosome 13 is at common frequencies in Africa & Southern Europe, Europe, and New Zealand, and co-localises with a SNP associated with resistance to the succinate dehydrogenase inhibitor (SDHI) pydiflumetofen. This TE is also located downstream of the gene ZymTr1|54560 of unknown function. *Family_1437* showed peak activity in the Iran and Israel populations, showing that the TE was inserted well before the application of fungicides in agriculture (Fig. 5). This supports the role of TEs as a source of standing genetic variation that can provide a host benefit much later in the species' evolution when novel challenges arise.

The DNA element (*family_152*) is found on chromosome 7 at a common frequency in Argentina & Uruguay and co-localises with 5 SNPs associated with resistance to the demethylase inhibitor (DMI) mefentrifluconazole. This TE is located downstream of gene ZymTr1|61488 (Proteasome assembly, GO:0043248) and upstream of gene ZymTr1|61489 (DNA-binding transcription factor activity GO:0000981 among other encoded functions). In contrast to *family_1437*, *family_152* shows a peak activity wave in the most recent Oceanian populations, and a smaller, more ancient activity wave in the North American populations (Figs. 2B, 5). This is consistent with a narrow time span between the emergence of adaptive TEs and strong selection pressures by DMIs potentially favouring the TE (Fig. 4).

Next, we determined whether the resistance-associated SNP allele correlated with the inserted TE. Only 16 isolates, all from the European population, have the resistance allele co-occurring with the presence of *family_1437* on chromosome 13 (Fig. 6A), whilst 146 isolates possess the resistance SNP allele without the TE, and 168 isolates possess the TE with the susceptible SNP allele. The majority of isolates, 1238, contain the susceptible SNP allele and lack the TE insertion. A more nuanced pattern is observed at the locus on chromosome 7, where the TE insertion is only present in isolates from Argentina & Uruguay, and all isolates with this TE (TE locus frequency = 50%, present in 14 isolates) have the resistance allele at the same three of five loci associated with resistance to mefentrifluconazole (positions: 1,456,821; 1,456,845; 1,464,109). Three isolates (11%) possess resistance alleles at these 3 positions in the absence of the LTR insertion, whilst the remaining 11 isolates (39%) in Argentina & Uruguay do not contain the LTR insertion or any of the resistance-associated SNPs (Fig. 6). The common frequency of this LTR insertion and a co-occurrence with 3 SNPs associated with fungicide resistance at this locus provides evidence to support a putative association between the candidate TE and local adaptation through modifying a fungicide resistance phenotype in the Argentina & Uruguay population, however functional characterisation will be required to confirm a mechanistic role for the TE insertion.

Throughout the global expansion of *Z. tritici*, populations have been exposed to heterogeneous selection driven by climate, including changes in temperature, precipitation, and seasonality. Recently, 21 copy number variants (CNVs) were found to be significantly associated with climatic factors at 14 independent loci[57]. We performed genome-wide environmental association mapping on TE loci and 10 principal components representing variation among 76 bioclimatic variables. We identified 2 TE loci significantly associated with variations in climate (Supplementary Data 10) including the same LINE located 1516 bp upstream of a putative effector on chromosome 3 (*family_270*) is also located 873 bp downstream of a *PIF-Harbinger* DNA TE (*family_997*) that is significantly associated with bioclimate PC4, with the top positive loadings associated with precipitation in the warmest and driest quarters, precipitation in the driest month, annual precipitation, and the minimum monthly climate moisture index (Supplementary Data 10, 11). On chromosome 15, an LTR element, *family_492*, is nested at a locus containing an unclassified TE (*family_1037*), which is significantly associated with bioclimate PC3, with the top positive loadings associated with precipitation in the wettest and coldest quarters, maximum monthly climate moisture index, annual precipitation, and net primary productivity (Supplementary Data 10, 11). However, in both cases, there is no co-occurrence of bioclimate-associated TEs with local adaptation TE candidates.

The closest polymorphic TE locus to an effector (Zt_3_904) is a LINE (*family_270*), found 1516 bp upstream of the transcription start site at common frequency in the Chile population (Supplementary Data 10). Zt_3_904 is a putative suppressor of flg22-induced reactive oxygen species (ROS) burst[58], with potential importance in successful host colonisation to overcome plant defences and maintain a redox environment conducive to cell viability[59]. 3 candidate TEs are associated with host genes within 3 separate biosynthetic gene clusters (BGCs) (Supplementary Data 10). A rolling circle element on chromosome 1 is associated with genes involved in terpene synthase synthesis, whilst a DNA TE insertion on chromosome 11 is associated with an NRPS-like BGC, and a MITE on chromosome 12 is associated with a non-alpha poly-amino acids (NAPAA) BGC. In all three cases, the genes have likely auxiliary BGC functions rather than being the core gene.

## Discussion

Despite growing evidence for TE polymorphism facilitating host adaptation, the TE dynamics leading up to the emergence of adaptive variation remain poorly understood. The extensive genomic resources for *Z. tritici*, combined with knowledge on the pathogen colonisation

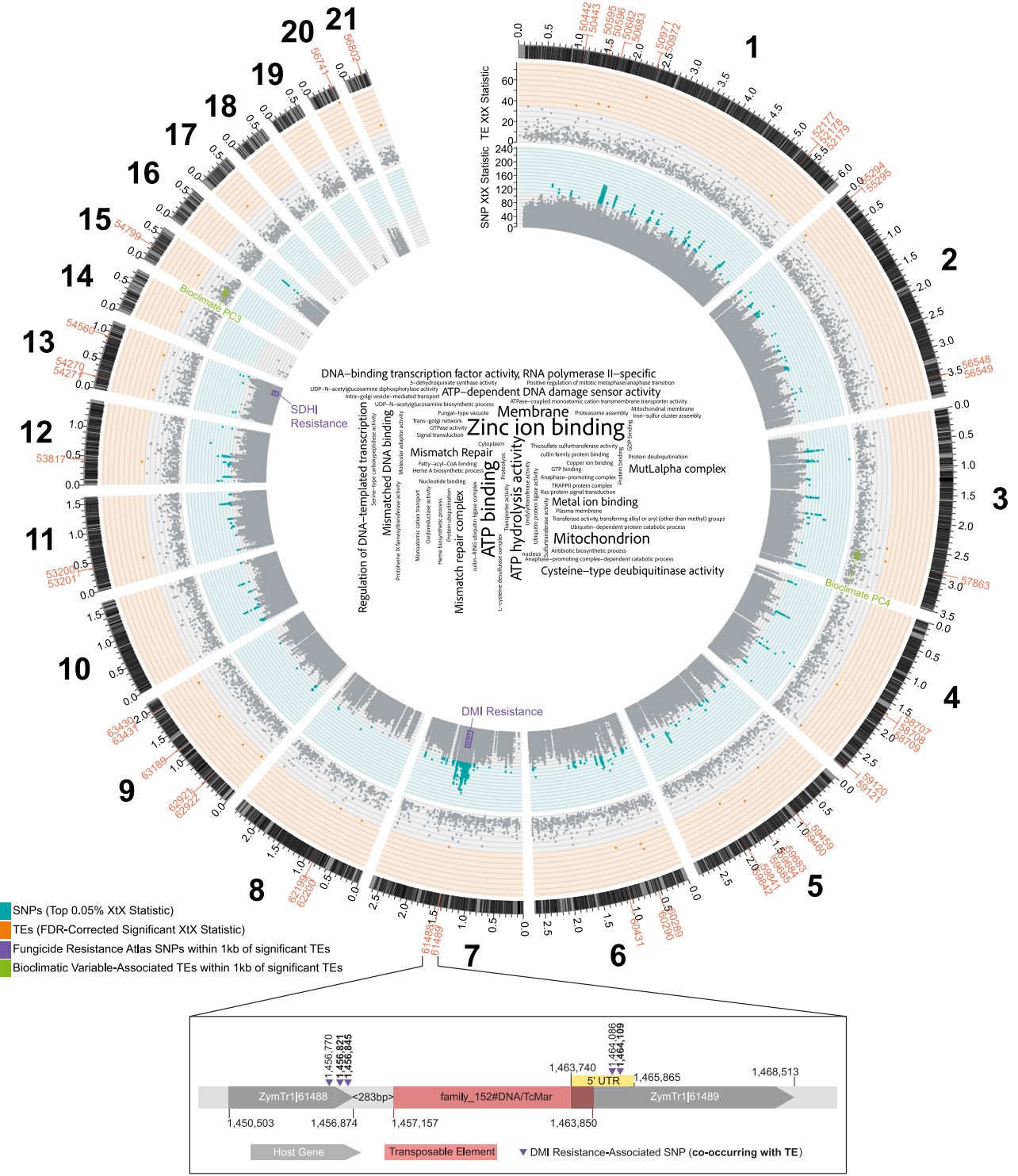

**Fig. 4 | Candidate adaptive TE loci detected by outlier search across the global pathogen distribution range.** Black bars on the outermost track show IPO323 gene annotations, with grey illustrating intergenic regions. Numbers indicate genomic coordinates in megabases (Mb). TE (outer track) and SNP (inner track) annotations mapping to chromosomes of the *Z. tritici* reference genome IPO323. Local adaptation candidates with significant XtX values are found in their respective shaded regions. Significantly differentiated SNPs have XtX values found in the blue shaded region of the inside track, with points coloured as indicated in the key. Significantly differentiated TEs have XtX values found in the orange shaded region of the outside track, with points coloured as indicated in the key. Genes within 1000 bp of candidate TEs are labelled outside the gene track in orange. SNPs implicated in fungicide resistance[46] and TEs associated with bioclimatic variables within 1000 bp of candidate overdifferentiated TEs are boxed and labelled, as indicated in the key. The wordcloud in the centre illustrates the GO terms assigned to genes nearby candidate TEs (highlighted in orange), with size proportional to the number of genes annotated with each GO term. GO terms for genes are also provided in Supplementary Data 9. Inset gene region schematic illustrates a single candidate TE and the host genome landscape surrounding this insertion on chromosome 7. Host genes and fungicide-resistance SNPs are labelled, as indicated in the key. SNPs in bold are found to colocalise with the candidate TE insertion in the Argentina and Uruguay population. Source data are provided as a Source Data file.

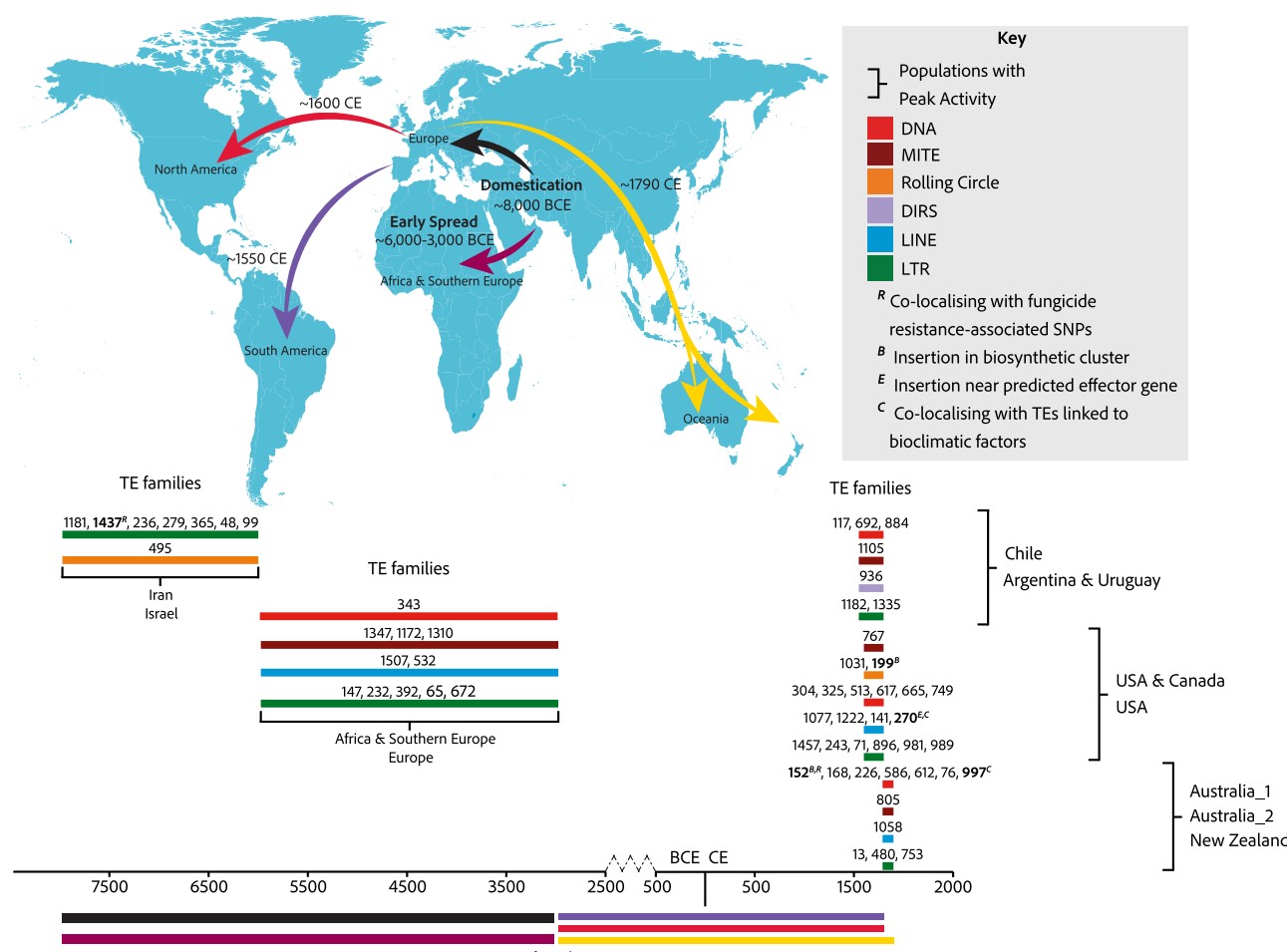

**Fig. 5 | Conceptualisation of adaptive TEs emerging through historic TE activation waves.** The map shows the global colonisation timeline of *Z. tritici* following wheat domestication. Bar width reflects the rough window of colonisation for the populations, with peak activity as shown in the key. Colours under the timeline correspond to those shown in the map arrows. TE families are shown by their family number corresponding to their names in the curated library. TE family peak activity is estimated based on the populations showing the highest mean copy number for TE families with a population mean TE copy number range >5. Candidate TE families associated with host functions are shown in bold, with host functions and TE classifications indicated in the key. Source data are provided as a Source Data file.

timeline over the last ~10,000 years, provided a unique opportunity to explore population-level TE-host evolutionary dynamics in the largest global sampling dataset to date. This framework enabled us to characterise population-specific waves of TE activity associated with the pathogen's global spread and facilitated the systematic characterisation of the dynamics underlying the emergence of TE-mediated adaptive variation. Curated TE databases are largely deficient in fungal sequences. As such, little is known about the diversity of TEs across the fungal kingdom despite the clinical and agricultural relevance of fungal pathogens and their TE-mediated adaptation.

We characterised TE activity across the colonisation timeline for *Z. tritici*, and found extensive population-specific activity, with accumulation of novel TE insertions coinciding with expansion into novel environments, putatively resulting from TE derepression in line with the longstanding stress-induced TE activation hypothesis[13,15,21,35,60–62]. Importantly, different colonisation stages are characterised by the proliferation of distinct TE families, and at different intensities, demonstrating the highly localised nature of TE activity among distinct populations. For example, the early spread of *Z. tritici* into Africa and Europe led to a wave of peak activity in 10 distinct TE families, whilst expansion into North America led to a wave of peak activity in 19 different TE families, including an intense increase in activity of the LTR retrotransposon *family_71*, which quadrupled in copy number

during this colonisation stage. Following increased TE family activity, a species may see the emergence of host benefits mediated by beneficial TE insertions. However, these two stages of TE impacts on the host genome remain poorly connected in nearly all systems studied to date. Among TE insertions with likely beneficial effects, we found the corresponding peak TE family activity to be predominantly in the last 500 years (8/11 TE families). This timeframe matches the pathogen's expansion into North and South America and Oceania. TE insertions likely impacted host functions such as fungicide resistance, BGCs, adaptation to the climate, and effector genes, suggesting a short time lag between the TE activity burst and adaptive significance. Conversely, the peak activity of the LTR retrotransposon, *family_1427*, dates back to the origin of the species in the centre of origin, but the association with fungicide resistance has only been created in recent decades, given the recency of such treatments in agriculture. This suggests that even older TE activation episodes can act as reservoirs for genomic variation of current adaptation processes.

Combining our findings, we propose that the expansion of *Z. tritici* into novel environments has resulted in TE family derepression in an environment-dependent manner. This localised TE activity, in combination with limited gene flow and the emergence of bottlenecks in founder populations, has resulted in significant levels of genomic differentiation among *Z. tritici* populations at the TE level. This variation

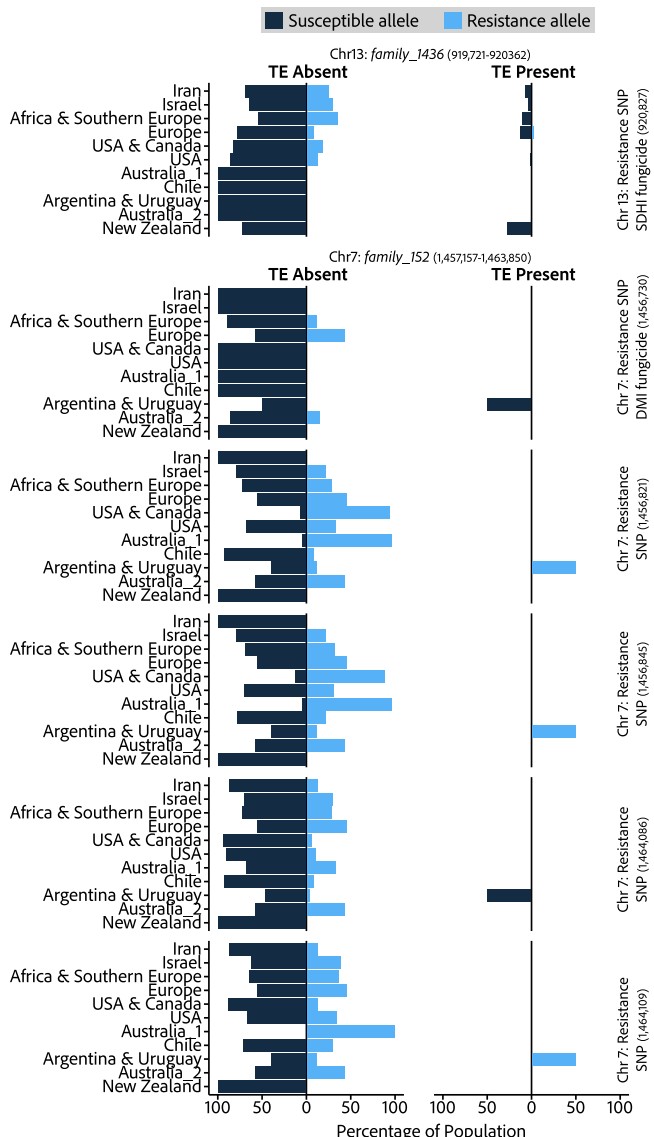

**Fig. 6 | Co-occurrence of fungicide resistance-associated loci and proximal to candidate adaptive TE insertions.** Populations are ordered using the colonisation timeline of wheat (Fig. 1). Each row is a distinct fungicide resistance-associated loci (SNPs), with bars indicating the presence of susceptible or resistance alleles, as indicated in the key. Left-hand plots show the proportion of isolates within a single population with susceptible or resistance SNPs in the absence of the proximal TE. Right-hand plots show the proportion of isolates within a single population with susceptible or resistance SNPs in the presence of the proximal TE. Source data are provided as a Source Data file.

TE insertion variants may encode regulatory or functional domains, expanding the repertoire of functional consequences for the genome. Previous work has focused on the role of SNPs in the rapid adaptation of *Z. tritici*[46,66–69], and the extent of TE activity observed argues for a shift in focus to incorporate TEs into future investigations to determine the genetic basis of rapid adaptation in pathogens such as *Z. tritici*.

Connecting adaptive TE insertions to the underlying TE activity in the genome reveals important features of TE activation lifecycles. Most TE families in *Z. tritici* are retained at low copy numbers, with a few families dominating the TE landscape. The LTR retrotransposon, *family_71*, shows a unique expansion of a magnitude not observed for any other families. This proliferation is consistent with the derepression of this TE family following the expansion of *Z. tritici* out of the Middle East, providing support for the role of stress in the induction of transposition activity[7,13,15]. Most research on TE dynamics in genomes only considers a subset of actual TE activity, as only TEs that successfully insert into host genomes are identified. Consequently, the host-TE interactions resulting in the success of this LTR family, whilst others persist at low levels, remain elusive, and we require an improved understanding of TE success rates to characterise the dynamics that result in either TE families being driven to extinction or TE families expanding in host genomes and contributing to host evolution. Whilst we cannot exclude the possibility that at least some of the observed changes in TE copy numbers among populations are the result of genetic drift and relaxed purifying selection, the most parsimonious explanation for our findings is ongoing TE activity.

In a gene-dense genome, the likelihood of a successful TE insertion is reduced, and detrimental effects on host genes are more likely. Hence, purifying selection on TEs should be more intense in gene-dense genomes. Furthermore, fixation of TEs in gene regions where they can exert regulatory effects on host genes is expected to provide a benefit to the host and be positively selected. Fixation resulting from inefficient purifying selection or genetic drift should also be less likely, given the large effective population sizes of the pathogen[33,34]. We identified 45 TEs that have risen to high population frequencies (≥10% and <95%) with significant signatures of local adaptation among populations. Because TEs can act as plug-and-play genomic units, they can provide host benefits by facilitating complex local adaptation. Such strong selection pressure to fix adaptive TE insertions might have been conferred by xenobiotic resistance[70], novel environments such as extreme cold and human-impacted landscapes[7,30], as well as host-pathogen interactions[15,71]. Consistent with this, we identified 2 candidate TE insertions associated with SNPs conferring resistance to fungicides, with a co-occurrence between a DNA TE and SNPs conferring resistance to mefentrifluconazole in the Argentina & Uruguay population. TE-mediated xenobiotic resistance mechanisms have been previously reported and functionally characterised in several insect lineages[72–76], and resistance to DMIs has been functionally demonstrated in *Penicillium digitatum*[77], which was conferred by MITE-altered expression of PdCYP51B. In *Z. tritici*, comprehensive antifungal resistance mapping efforts revealed that resistance loci were over-represented in repetitive regions of the genome, suggesting a broad involvement of TE-driven adaptive variants[46,66]. Given the strong selection pressure imposed on plant pathogen populations by fungicides and the ability of TEs to exert regulatory effects on nearby host genes, TE insertion variants should be the primary focus for the detection of emerging resistance in the field.

The advances in population-level sequencing of TE variants and connecting such polymorphism to adaptive trait variation raise major questions about how much of the total adaptive variation in species gene pools has been generated by TE activity alone. Quantifying these contributions requires characterisation of TE dynamics beyond traditional model systems and species groups. The approach taken here illustrates the widespread and repeated contributions of TEs to the

can then provide a host benefit through (i) contributing to extant levels of standing genetic variation, facilitating adaptation in derived populations, or (ii) providing pre-adaptation genetic variation to facilitate rapid adaptation thousands of generations later. Supporting this, we observe massive increases in TE activity over timescales as short as 25 years. To our knowledge, increases in TE copy number of this magnitude (mean TE copy number increase = 387) over such a short timescale have not previously been observed in natural populations. Assuming a generation time of one year, in the North American population, we observe a mean transposition rate of 15.48 new insertions per generation, whilst ~0.0040–0.40 SNPs per generation are expected to arise at a neutral mutation rate of $10^{-8}$–$10^{-10}$ substitutions per base pair per generation[63–65]. Therefore, the potential of TEs to generate new variants is magnitudes higher than SNPs. Furthermore,

generation of genomic novelty within a single species in the under-studied fungal kingdom. We shed light on the power of TEs to generate extensive genomic variation within and among populations over extremely short timescales, measured in decades rather than millions of years. We find that exposure to novel environments and the establishment of new populations underpins distinct waves of TE activity, generating adaptive genetic variation and geographic differentiation. These waves of TE activity result in large numbers of new insertions even in the suppressive landscape of a gene-dense genome. The dynamism of TE activity may overcome limitations in species gene pools and drive responses to changing environments more quickly than previously expected. Our findings support tight integration of TE variant monitoring into investigations of rapid adaptation, particularly in systems where TEs might be derepressed. The activation of TEs in response to novel environments also sheds light on the potential impact that climate change could have on host genomes, particularly related to the movement and expansion of pathogens relevant for food security and human health. However, routine implementation of long-read sequencing in population studies will be required to fully resolve TE dynamics. Beyond this, future research needs to broaden the sampling of TE diversity beyond traditionally studied species groups.

## Methods

### Generation of a manually curated TE library for *Zymoseptoria tritici*

We annotated 19 reference-quality genome assemblies representing the global diversity of *Z. tritici* and 4 reference-quality genome assemblies for the sister species (*Z. pseudotritici, Z. passerinii, Z. ardabiliae*, and *Z. brevis*)[78–81] with Earl Grey (v3.0; https://github.com/TobyBaril/EarlGrey)[47] to de novo detect and generate putative TE consensus sequences. TE consensus sequences were first clustered using CD-Hit-Est (v4.8.1)[82,83] to reduce redundancy in the initial library by grouping sequences with 90% similarity across at least 80% of the total length of the longer sequence to prevent collapsing of chimeric consensus sequences. We manually curated consensus sequences following the protocols described in ref. 84. Genomic copies of each TE with 1000 bp of flanking sequence at each end were obtained from all 23 genome assemblies using a BLAST, Extract, Align, Trim process[85]. We selected the 25 longest hits, along with up to 75 random hits for each TE query sequence, and generated multiple sequence alignments with MAFFT --auto (v7.505)[86] followed by T-COFFEE (v13.45.0.4846264)[87] to remove columns composed of ≥80% gaps. Resultant sequence alignments were manually curated to define boundaries of single-copy DNA sequence (i.e., visible TE boundaries), identify TE subfamilies, and remove regions of low conservation. New majority-rule consensus sequences were constructed with EMBOSS (v6.6.0) cons[88] after which TE consensi were visually inspected using TE-Aid (https://github.com/clemgoub/TE-Aid/) to identify diagnostic features characteristic of the main TE classifications[89]. We recorded the presence of TIRs, along with their sequence, and homology to curated TEs in Dfam (v3.7)[90,91] using nhmmscan (HMMER v3.3.2)[92]. These sources of information were combined to manually classify each TE consensus sequence using the convention 'ZymTri_2023_family_n#classification/family' for compatibility with RepeatMasker. Consensus sequences classified with low confidence are labelled with both '?' and '_LowConf' appended to the name. Following de novo TE extension, curation, and new consensus generation, a second redundancy filter for the final TE library was run in CD-Hit-Est to collapse TEs to family-level using the 80-80-80 rule (≥80% identity over ≥80% sequence length over ≥80 bp)[84,89]. CD-Hit-Est alignments were manually inspected to select the representative sequence in each cluster by choosing the most intact sequence with the highest classification confidence level. Chimeric sequences erroneously clustered with their individual counterparts were manually separated to retain sequences for the chimeric TE and their individual elements.

In total, 331 TE consensus sequences are found in the *Z. tritici* library, of which 199 are confidently classified, 27 are classified with low confidence, and 105 are putative TEs labelled as Unclassified. The manually curated TE library can be freely accessed at https://doi.org/10.5281/zenodo.8379981[48,93].

The non-redundant reference-quality genome assembly was annotated using the manually curated TE library using earlGreyAnnotationOnly in the Earl Grey package, which annotates TEs using RepeatMasker, before post-processing annotations to resolve fragmented and overlapping annotations, and to identify full-length LTR elements[47]. Annotations are provided in Supplementary Data 2 and are used for the McClintock2 analysis described below.

### Sourcing of short-read genomes for the global panel

We obtained short-read genomes provided in refs. 42, 46. Samples cover the global distribution of *Z. tritici*, from the centre of origin in the Middle East to the most recently derived populations in Oceania. Samples were taken over a span of decades, with the earliest samples isolated in 1981 and the latest in 2019. Genome information is available in Supplementary Data 12.

### SNP calling

We generated a non-redundant genome assembly representing the global diversity of *Z. tritici* from the reference-quality *Z. tritici* genome assemblies using REVEAL (REcursive Exact-matching ALigner) (https://github.com/jasperlinthorst/reveal), starting with the IPO323 assembly and iteratively aligning subsequent *Z. tritici* genome assemblies. The finished draft genome was made to conform to the layout of the initial backbone sequence of IPO323 (finish –order=chains). Unmapped fragments of at least 10 kb were appended to the non-redundant genome assembly fasta with headers informative of their genomic locations in their respective source genomes.

We performed variant calling following the protocols and using scripts described in ref. 42. Briefly, sequence variants were called using short-read mapping to the non-redundant genome assembly representing the global diversity of *Z. tritici*. Reads were filtered and trimmed using Trimmomatic (v0.40)[94] to remove adaptor sequences, trimming leading and trailing bases with a quality lower than 15, and removing sequences shorter than 50 bp. Trimmed reads were mapped to the non-redundant genome assembly using bowtie2 (v2.5.2)[95,96]. GATK4 (v4.4.0.0)[97] was used to call short variants with the commands HaplotypeCaller, Combine GVCFs, and GenotypeGVCFs, setting ploidy to 1 and the maximum number of alternative alleles to 2.

We performed several steps of quality filtering, starting with a standard set of hard filters using the GATK quality metrics with thresholds set using visual inspection of the metrics across the called variants. The per-site filters used were $FS > 10$, $MQ < 20$, $QD < 20$, ReadPosRankSum, MQRankSum, and BaseQRankSum between $-2$ and 2. Genotypes with depth <3 were also removed.

Genotyping errors can occur due to misalignment or the presence of repeated sequences in the genome. In the case of *Z. tritici*, erroneous variant calls can arise where near-equal numbers of reads support the reference and alternative allele, which should be recognised as errors in a haploid organism. To resolve these errors, genotypes with <90% of reads supporting the called allele were discarded. Following this, included samples were filtered with vcftools (v0.1.16)[98] –missing-indv and –depth options to remove samples with >20% missing data and a mean depth of coverage <6 on core chromosomes of the IPO323 backbone of the non-redundant genome assembly. We removed clonal, or near-clonal, samples by creating a network of isolates with an identity-by-state value >0.99 as calculated by plink (v1.90b6.21)[99]. Subgraphs representing groups of clones were extracted in R (v4.3.3)[100] using the tidygraph and ggraph packages[101,102] in the RStudio

IDE[103,104]. For each network, the isolate with the lowest amount of missing data was retained, whilst all other isolates were discarded. Following these filtering steps, the final isolate count was 1953, and the total variant count (indels and SNPS) was 11,066,828.

## Population structure analyses

To estimate the global population structure of *Z. tritici*, the total SNP set was filtered to sample 1 SNP/kb, with no missing data, and a minor allele frequency of 0.05, resulting in a total of 10,704 SNPs. A clustering analysis was run using the snmf clustering method in the LEA R package[105]. Clustering was run for K values (number of clusters) ranging from 1 to 15 with 10 repetitions per K. The best K was calculated using the entropy method in snmf to determine the best-fit model, in addition to manual evaluation based on the smallest cluster size and the number of isolates assigned to a cluster with a coefficient of >0.75. For downstream analysis, isolates were assigned to discrete populations at the coefficient threshold at K = 13. We collapsed two clusters predominantly assigned to isolates of European origin into a single Europe cluster, in line with previous studies, which assign European isolates to a single dominant cluster[42]. This resulted in a total of 12 distinct genetic clusters, including one labelled as Hybrid, containing isolates that could not be assigned at the coefficient threshold.

## Benchmarking short-read transposable element annotation tools

Several tools exist for annotating TEs from short-read sequencing to facilitate population-level studies. 12 tools are included in the McClintock2 meta-pipeline (https://github.com/bergmanlab/mcclintock)[49], namely: ngs_te_mapper[106], ngs_te_mapper2[107], PoPoolationTE[108], PoPoolationTE2[109], RelocaTE[110], RelocaTE2[111], RetroSeq[112], TEBreak[113], TEFLoN[50], TE-locate[114], TEMP[115], and TEMP2[52]. We were unable to successfully run relocaTE2, and therefore, we evaluated the performance of 11 methodologies to determine which should be used to detect TE insertions in our genome panel. A combination of tools is potentially required to detect both reference and non-reference TE insertions from short-read datasets. In this context, reference insertions are those present in the reference genome (i.e., the assembly to which short reads are mapped), whilst non-reference insertions are those that are present in the short reads but missing in the reference genome. To determine the best tool, or combination of tools, to annotate TEs across 1953 genomes, we employed several benchmarking analyses for method selection. We consistently ran all methods in McClintock2 with default parameters.

To assess tool annotation performance (analysis 1), we used chromosome 1 of the CRI10 *Z. tritici* reference genome[78] as the reference assembly, with TEs annotated by RepeatMasker (-s -norna) (v4.1.4)[116] using our de novo TE library. 100x depth short reads were generated using ART (version MountRainier-2016-06-05)[117] (-ss HS25 --noALN -l 100 --paired --mflen 375 --sdev 110 --fcov 100) and a modified version of chromosome 1 of CRI10 with 36 extra random TE sequences inserted at known positions. The aim of analysis 1 is for each tool to successfully annotate all reference TEs (i.e., those present in the unmodified chromosome 1 CRI10 reference) along with 36 non-reference TEs (i.e those present in the short reads from modified CRI10, but not in the reference assembly to which reads are being mapped).

To assess the ability of tools to avoid false positive annotations (analysis 2), we used the modified chromosome 1 of CRI10 (i.e., with the extra 36 random TE sequences added) as the reference assembly, with TEs annotated by RepeatMasker using our de novo TE library. 100x depth short reads were generated using ART (-ss HS25 --noALN -l 100 --paired --mflen 375 --sdev 110 --fcov 100) and the unmodified version of chromosome 1 of CRI10. The aim of analysis 2 is for each tool to successfully annotate only reference TEs, but to miss the 36 artificially added TE insertions, which are not present in the short-read

dataset. Here, any TEs labelled as non-reference are false positives or mislabelled if they are indeed reference insertions.

To assess the consistency between TE polymorphism calls from short-read and long-read data (analyses 3 and 4), we used the IPO323 genome assembly as the reference assembly, with TEs annotated by RepeatMasker using our de novo TE library. For analysis 3, McClintock2 methods were used to map short reads from the CRI10 isolate[42] to annotate TEs. For analysis 4, pbsv (v2.9.0) (https://github.com/PacificBiosciences/pbsv) was used to call structural variation (SV) using long reads from the same CRI10 isolate[78]. The aim of analyses 3 and 4 is to use both long and short reads from the same isolate (i.e., genetically identical read sets) to determine whether SV calls using long-read data support TE absence (i.e., deletions in comparison to the reference) and non-reference TE presence (i.e., TEs present in CRI10 but absent in IPO323).

For analyses 1 and 2, reference TE loci were intersected with the TE annotations GFF3 file produced by each method in McClintock2 using intersectBed in BEDTools (v2.31.0)[118]. True positives are defined as shared hits between the reference and method GFF3 files. False negatives are defined as annotations in the reference GFF3 missing in the method GFF3. False positives are defined as annotations present in the method GFF3 but missing in the reference GFF3. Method annotations were labelled based on the TE classification and insertion type (i.e., reference or non-reference TE) to evaluate overall method performance using R. Pearson's Chi-Squared Test for Independence was performed in R (chisq.test) to test for significant differences among TE annotation methods.

For analyses 3 and 4, TE loci annotations generated by McClintock2 methods in analysis 3 were intersected with the SV calls generated by pbsv in analysis 4 using intersectBed in BEDTools. Shared and unique loci were analysed in R, and labels were assigned to evaluate the level of evidence supporting a call from the short-read analyses. An annotation using a McClintock2 method was labelled as supported by long-read data if the McClintock2 annotation is a non-reference insertion, and the SV call is an insertion. Additionally, a reference annotation using McClintock2 methods was labelled as supported if there was no SV call, as reference insertions should not generate SV calls, as they are present in both the reference genome (i.e., IPO323) and the long-read dataset (CRI10 reads). McClintock2 method calls were labelled as dubious if: (i) the McClintock2 call is a non-reference insertion, and the SV call is imprecise; (ii) the McClintock2 call is a reference insertion, and the SV call is imprecise, deletion, or insertion, as TE insertions are likely to have degraded to different extents between CRI10 and IPO323. McClintock2 method calls were labelled as unsupported if: (i) the McClintock2 call is a non-reference insertion, and the SV call is deletion, translocation, inversion, or duplication; (ii) the McClintock2 call is a reference insertion, and the SV call is inversion, translocation, or duplication; (iii) the McClintock2 method call is a non-reference insertion, and it is missing from the SV calls. Pearson's Chi-Squared Test for Independence was performed in R (chisq.test) to test for significant differences among TE annotation methods.

Following benchmarking, we decided to combine three methods: TEFLoN[50], PoPoolationTE[51], and TEMP2[52], with the latter used only to annotate non-reference insertions. The non-redundant set of annotations from this combination of methods was benchmarked using analyses 1 to 4 to assess the improvement gained by combining methods. For all downstream analyses, we use this combination of methods, and a non-redundant set of TE annotations is generated using intersectBed in BEDTools and custom scripts to add PoPoolationTE and TEMP2 non-reference annotations to TEFLoN annotations. Full benchmarking results are provided in Supplementary Data 1.

Before annotating the *Z. tritici* global panel, we wanted to understand the potential impact of variation in sequencing technology and read depth on our ability to detect TEs. We took the isolate with

the most reads in each collection and downsampled these to read thresholds calculated based on the isolate from the collection with the lowest read count and shortest reads (Hartmann_FstQst_2015). All test isolate read sets were downsampled to the following read count thresholds using seqtk sample (v1.4-r122) (https://github.com/lh3/seqtk) with randomly specified seeds per paired-end read set: (i) 9,860,000; (ii) 8,410,000; (iii) 7,480,000; (iv) 6,540,000; (v) 5,610,000; (vi) 4,670,000; (vii) 3,740,000; (viii) 2,800,000; (ix) 1,870,000; (x) 935,000. Models were fitted between read count and TE locus count in R, and the minimum threshold for each dataset was determined as the minimum number of reads before the change in gradient was ≥0.1. Each isolate dataset was labelled as accepted if the read count was above this threshold, borderline if the read count was within 10% of the threshold and rejected if the read count was >10% below the threshold. In total, 418 isolates were accepted, 104 were borderline, and 1431 were rejected. Given the large number of rejected isolates, we retained all isolates in the global panel for downstream analyses. Consequently, we likely underestimate the TE content in rejected isolates and therefore will only detect the strongest differences in TE locus frequency among populations and miss more subtle and potentially more recent patterns.

### Annotating transposable element insertions in the global panel

Following benchmarking, we analysed all 1953 short-read genomes with McClintock2 methods TEFLoN, PoPoolationTE, and TEMP2 (non-reference only) to annotate TEs in each isolate. Here, each set of sequencing reads was mapped to the non-redundant assembly representing the global diversity of *Z. tritici* generated for this study (see "Methods: SNP calling"). McClintock2 was also provided with the manually curated TE library for *Z. tritici* in FASTA format, a GFF3 file containing the TE annotations for the non-redundant assembly, and a taxonomy file providing mapping between each unique TE in the GFF3 file and its respective classification. For each run, the resultant TE annotations were combined to make a non-redundant set comprising positions identified by TEFLoN, PoPoolationTE, and TEMP2, using BEDTools to add unique positions from PoPoolationTE and TEMP2 to the TEFLoN annotations.

By default, McClintock2 filters annotations for quality. For a TE annotation to be called, all three methods require all TE annotations to be supported by ≥10% of the total reads, whilst TEFLoN also requires a minimum of 3 reads supporting insertion presence and non-reference hits to be supported by both 5' and 3' breakpoints. PoPoolationTE requires insertions to be supported by reads on both the forward and reverse strand, whilst TEMP2 requires insertions to be supported by reads mapping to both ends of the insertion. We visualised the frequency of reads supporting each TE annotation using R and ggplot2 (v3.5.0)[119] to determine if further quality filtering was needed. Given that *Z. tritici* is haploid, we set a further quality threshold to remove any positions with a read frequency ≤0.5, equivalent to a TE annotation having a 50% chance of being false. For each isolate, ~90% of total TE annotations were supported by ≥90% of mapped reads supporting the presence of a TE insertion.

TE locus information was parsed in R to label every locus from each isolate. Each locus was given an individual ID, and loci were defined as shared among isolates if annotations were for the same TE family on the same contig, at overlapping coordinates and/or within 100 bp of each other. Each locus was labelled with diagnostic information: (i) a unique numeric locus identifier; (ii) as reference if it was present in the non-redundant genome assembly, or non-reference if it was absent; (iii) the contig where the annotation is found; (iv) the TE family identifier from the de novo TE library; (v) Either rare if the TE is present in <10% of the isolates in the respective population, common if the TE is present in ≥10% and <95% of the isolates in the respective population, and fixed if the TE is present in >95% of the isolates in the respective population, following the recommendations of ref. 53.

To determine the importance of TEs in driving population differentiation, we generated a TE locus presence/absence matrix. TE loci were filtered to remove positions present in <10 isolates, reducing the locus count to 17,129. A principal component analysis (PCA) was performed using prcomp in R, and the results were visualised with ggplot2. All subsequent downstream analyses were performed using R, in the RStudio IDE, and the tidyverse (v2.0.0), data.table (v1.14.10), and magrittr (v2.0.3) packages. Plots were generated with ggplot2[119] and KaryoploteR[120].

### Detection of putatively adaptive transposable element insertions

We performed genome-wide scans for adaptive differentiation using the core model in BayPass (v2.4)[54], which accounts for the correlation structure among allele frequencies and facilitates the detection of putative TEs subject to adaptive variation based on the XtX statistic[121]. The core model has several advantages over other methods of detecting adaptive variation as it explicitly accounts for the demographic history of populations, accounting for co-variation of allele frequencies due to shared history and genetic drift (covariance matrix Ω), whilst making no assumptions about the underlying demography. BayPass calculates the XtX statistic, which allows consideration of more complex population histories (i.e., migration and ancestral admixture), and is corrected for the scaled covariance of population allele frequencies[54]. Significant outlier XtX values are used to identify variants with strong signatures of adaptive differentiation. BayPass exhibits a low false discovery rate and high power for strongly associated variants, but is more limited in its ability to identify more weakly associated variants[54], therefore limiting our detection to strongly associated variants.

Only isolates that were confidently assigned to a genetic cluster were included, resulting in a dataset containing 1569 individuals. We filtered TE loci to maximise our ability to detect putatively adaptive loci with high confidence: (i) TE loci unique to a single isolate were removed; (ii) TE loci were removed if they were not found in at least 10% of the isolates of at least one population (i.e., TE loci rising to common frequency in at least one population were retained); (iii) TE loci were removed if they were not polymorphic in at least one population. This reduced our analysis to 11,073 TE loci. The core model was provided with the relatedness matrix used to assign isolates to their respective genetic clusters (see "Methods: Population structure analyses"), to ensure consistency in the estimations of population structure. The genotyping input file was generated from the TE presence/absence matrix used above in R. Briefly, the genotyping file contained TE presence and absence counts (in lieu of reference and alternative allele counts if SNPs were used) per site and per population. TE presence counts were used as the alternative count, whilst TE absence counts were used as the reference count[122].

The BayPass core model was run with 40 pilot runs of 1000 iterations, followed by a burn-in of 1,000,000 iterations and 10,000,000 model runs sampled with a thinning interval of 1000 to result in a total value count of 10,000. The model was run three times with different seeds to verify convergence among separate model estimates. Following model runs, TEs with very low allele frequencies (MAF < 0.01) based on the mean of the posterior distribution of the frequency of the reference allele across populations for each site were filtered, as recommended in the BayPass documentation (https://forgemia.inra.fr/mathieu.gautier/baypass_public/)[54]. To control for multiple testing, the *p*-value distributions were manually inspected. As the *p*-value distributions were uniform, we applied a false discovery rate correction using the Benjamini-Hochberg method in R, as recommended in the BayPass documentation. We focused on XtX statistic scores below an FDR-corrected significance threshold of 0.05 and define these as significant outliers and putative candidate TE loci under positive selection within a population. Loci were then filtered

further so that only TEs on the IPO323 chromosomal-level backbone of the non-redundant genome assembly were considered, as these contained genomic sufficient context along with gene annotations to facilitate downstream analyses.

We also performed scans using the same parameters and three long model runs using SNP loci to assess the co-localisation among overdifferentiated TE and SNP loci. Similarly, only isolates assigned to genetic clusters were included (1569 individuals). Loci with >10% missing data were removed, resulting in 4,189,176 SNPs. The genotyping input file was generated from the SNP-calling VCF in R, resulting in a file containing SNP allele calls per site and per population. Consistent with the TE analysis, SNPs with very low allele frequencies (MAF < 0.01) based on the mean of the posterior distribution of the frequency of the reference allele across populations for each site were filtered. *P*-value distributions were manually inspected, and a q-value transformation was applied to control for multiple testing in R using the qvalue package(v2.36.0)[123], as recommended in the BayPass documentation. SNPs were defined as significant outliers if their XtX statistics were in the top 0.05% of values, and their associated q-values were below 0.05.

### Integration of bioclimatic, fungicide resistance and gene function datasets

To gain insight into the potential mechanistic impacts of candidate TE loci, we obtained: (i) Gene annotations for the reference assembly IPO323[124]; (ii) SNP loci significantly associated with fungicide resistance from a *Z. tritici* fungicide resistance atlas[46]; (iii) Data for 76 bioclimatic variables from the Climatologies at high resolution for the earth's land surface areas dataset (CHELSA v2.1) (https://chelsa-climate.org/bioclim/)[125,126]; (iv) Effector gene amino acid sequences from[58] (v) Biosynthetic cluster annotations for the reference assembly IPO323[57].

Genes within 1000 bp of candidate TE loci were obtained using BEDTools window. Candidate gene amino acid sequences were annotated with InterProScan (v5.67-99.0)[55,56] to assign predicted gene functions, GO terms, and MetaCyc and Reactome pathway memberships.

SNPs with significant associations to fungicide resistance within 1000 bp of candidate TE loci were obtained with BEDTools window. Information on the specific fungicides to which resistance is conferred and SNPeff predicted impacts were obtained from the fungicide resistance atlas[46].

For each isolate, geographical coordinates were obtained from the attached metadata. In some cases, assigned coordinates were inferred based on the most precise sampling data available, as previously described[42]. The sampling coordinates for each isolate were used to approximate the environmental conditions from all 76 bioclimatic variables, which are averaged over the period 1981–2010[125,126]. We generated a pseudo-VCF containing all TE calls for each isolate, where 0 represented TE absence and 1 represented TE presence (Supplementary Data 3). We identified genotype-to-environment associations among TE loci and bioclimatic variables using vcf2gwas (https://github.com/frankvogt/vcf2gwas)[127], specifying the use of univariate linear mixed models (-lmm). Given the high correlation among distinct bioclimatic variables (Supplementary Fig. S6), we also performed PCA-based dimensionality reduction on the bioclimatic variables within vcf2gwas (--PCA 10), which were then used as the phenotypes for the genotype-to-environment association. Following this, TEs with significant genotype-to-environment associations were filtered to retain those within 1000 bp of candidate TE loci.

We obtained predicted effector gene amino acid sequences[58] and identified corresponding loci in the IPO323 reference assembly used in this study using tblastn[85] followed by manual curation (Supplementary Data 13). A bed file was generated, and BEDTools window was used to identify putative effector genes within 1000 bp of candidate TE loci.

Biosynthetic gene clusters (BGCs) are arrays of genes involved in the production of diverse secondary metabolites[128], and they often exhibit large presence/absence variation among individuals and populations[57]. We obtained BGC annotations for the IPO323 reference assembly from ref. 57, and cross-referenced the gene members with genes located with 1000 bp of candidate TEs to assign predicted BGC functions to genes at these loci.

Plots to illustrate the association among candidate TEs and aforementioned data sources were generated using circos (v0.69-8)[129].

### Reporting summary
Further information on research design is available in the Nature Portfolio Reporting Summary linked to this article.

## Data availability
Supplementary Data files 1 and 4–14 are available in the article. Supplementary Data 2 and 3 have been deposited in the Zenodo database under https://doi.org/10.5281/zenodo.17189961 (https://doi.org/10.5281/zenodo.17189961). The manually curated TE library has been deposited in the Zenodo database under https://doi.org/10.5281/zenodo.8379980 (https://doi.org/10.5281/zenodo.8379980) and has been submitted to the Dfam consortium (Dfam Release 3.9). Source data for figures are provided in the Source Data archive on Zenodo under https://doi.org/10.5281/zenodo.17189961 (https://doi.org/10.5281/zenodo.17189961). All sequencing data are available from the NCBI Sequence Read Archive (https://www.ncbi.nlm.nih.gov/sra). Individual accession numbers can be retrieved from Supplementary Data 12. Climatic data were obtained from the publicly available WorldClim database version 2.1 (https://worldclim.org).

## Code availability
All custom code used in this study is provided in a public GitHub repository at https://github.com/TobyBaril/Baril2025_ZT_TE_PopGen and under https://doi.org/10.5281/zenodo.17021596 (https://doi.org/10.5281/zenodo.17021596)[130].

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

## Acknowledgements

D.C. was supported by the Swiss National Science Foundation (grant 201149) and by an Innosuisse grant (32532.1 IP-LS).

## Author contributions

T.B. and D.C. conceived and coordinated the study. T.B. performed analyses. G.P. provided datasets. D.C. provided funding and supervised the work. T.B. and D.C. wrote the manuscript with input from G.P.

## Competing interests

The authors declare no competing interests.
