## [Transparent Peer Review file · Nature Communications]

Historic transposon mobilisation waves create distinct pools of adaptive variants in a major crop pathogen

Corresponding Author: Professor Daniel Croll

Version 0:

Reviewer comments:

Reviewer #1

(Remarks to the Author)

Summary:

This study uses a large-scale dataset of genome sequences from thousands of fungal isolates to investigate the insertion dynamics of TEs during the species' colonization of globally widespread regions. Overall, the study is impressive in scope — thousands of samples and many TEs — with a clear, impactful headline detecting population-specific TE burst and potentially adaptive TE insertions. I have specific suggestions that I hope will improve the study and enhance the clarity of presentation. Overall I am supportive of publication.

Major:

It is somewhat surprising that after manual curation, 105/331 (>30%) of Zymoseptoria TE models remain unclassified. Can the authors comment on this? Do the authors suspect that Zymoseptoria harbors TEs that mobilize through currently uncharacterized mechanisms?

The authors should also provide the percent of the genome occupied by TEs in each class for each assembly used for TE model curation. This metric is useful because it is not affected by fragmentation issues in TE annotations, which can show biases across element types. Metrics such as “copy number” unfortunately suffer from this bias. Relatedly, occasional fragmented TE annotations may introduce redundant TE presence/absence calls. Can the authors comment on whether or not there is evidence of this in their data, and if so, how they address this challenge?

Minor:

Transposable elements should be defined earlier in the introduction. They are currently mentioned first in the first paragraph and defined in the second paragraph

Line 185: While I do not see a problem with separating DNA TEs and MITEs into separate groups throughout the MS, the authors should at least clearly define the relationship between them. I am assuming that MITEs refer to nonautonomous “DNA TEs.”

Some of the initial results go on a bit long describing the dataset and proximal results such as TE counts without getting to the main points of the paper. I suggest reducing some of the descriptive parts to supplemental tables.

Line 392: Start sentence with “Three” rather than “3”

I suggest reducing the amount of passive voice used throughout the methods to enhance readability.

Reviewer #2

(Remarks to the Author)

Baril et al., characterize TEs in >1900 publicly available genomes of *Z. tritici*, an important wheat pathogen. They identified

>3million TE insertions. They claim that TE activity increased with the habitat expansion of the pathogen, and that this increased activity led to adaptive TE insertions. They have interesting data and this was clearly a massive effort, but I do not think that the data are strongly supporting any of these two claims. I do not see convincing evidence that TE activity increased during the last 500 years nor that any of the insertions have contributed to adaptation.

major

- where do I see that TE activity increased with global expansion of the pathogen? I guess in Figure 1. But upon comparing TE copy numbers in the ancestral populations (Iran and Israel) and the novel populations (Africa, America, Australia) the copy numbers seem very similar. Except perhaps in Australia where copy number is lower but this is contrary to their claim. The magnitude of the copy number differences seems in any case very small.
- 156 related to the above; they mention that copy numbers increased significantly but no p-value is provided (at least not in the main manuscript). Even if this is significant also the magnitude of the change should be provided; with very large data sets (1900 samples) even minor changes may be significant. As mentioned above the copy number increase seems very minor (except perhaps for LTR family_71)
- 161 they claim the data is consistent with increased activity, but where is evidence for this. an increase in TE copy number could also be due to founder effects, relaxed purifying selection etc, so there are several alternative hypothesis to consider for differences in TE copy numbers (which i do not see anyhow)
- 189 'Beyond changes in TE activity..' again I do not see evidence that the activity changed (could all be founder effects, drift...)
- 193-194 this is interesting; sub-populations from the USA sampled before 1990 contain <2000 TEs whereas subpopulations sampled in 2010 contain more than 2000 TEs. so this could be evidence that TE activity increased recently. but with short read data it could also be due to problems with the data or the analysis, see this important paper <https://elifesciences.org/articles/28297> As a creative idea they used immobile elements (exons) to test their pipeline. It would be important if the authors also adapted this control. If their pipeline shows that the number of immobile elements (eg exons) is similar for the samples <1990 and 2010 but the numbers of TEs differs, then that would support increased activity in the USA between 1990 and 2010 (but than I do not get why activity just increased in the last decades, NA was colonized by *Z.triticia* around 1600 according to their fig1).
- Fig2A it is quite unclear to me what 'affinity' means? as far as i guess it means overrepresentation or underrepresentation of TE families in the different *z.tritici* strains?
- I think it should be made more clear early in the manuscript that the work is based on previously published data. It became clear to me only by looking at the material and methods.
- they claim they did a pangenome approach. which suggests to me, and perhaps to others a pangenome graph. But as far as i can tell the authors just did a de novo annotated TEs with EarlGrey in multiple related species and they call this pangenome approach, which is very misleading. This is overselling, especially given that people will likely associate this with pangenome graphs (e.g. using long-read assemblies) which indeed allows for a quite robust identification of TE polymorphism. But in contrast to this, their analysis is based on 1900 short-read data which allows for a far less robust identification of TEs insertions. Although I think they did a good job benchmarking the different tools for analysing the short read data.
- 258 the majority of the insertions occurred during the last 500 years. Where is the evidence for this? They did not calculate the age of any TE insertion. It is possible to compute the age of the insertions eg by LTR divergence or by the age of the haploblock around a TE, but they did not do this.
- Fig3 is interesting, but why are there so many TE loci just found in Europe?. Is this just because most samples are from Europe? What do we learn from this graph other than that most samples are from Europe and the 2nd most from the USA (though in a convoluted way)? I think it would be important to normalise the TE loci count by the number of samples from each region.
- how were adaptive TEs identified? just with BayPass? but what is this tool doing, what kind of information is it utilizing to infer positive selection. Just mentioning BayPass is not sufficient for such a central message of this work. It needs to be very transparent how positively selected loci were identified, and what is the false positive rate of BayPass and to what kind of problems it is susceptible (e.g. population structure, sequencing errors). How was BayPass validated for this work? Is BayPass just using differences in the insertion frequencies among populations? Please consider alternative explanations such as founder effects and drift and how this could impact selection scans with BayPass.
- 280,286 etc adaptive TE loci: please write putatively adaptive, there is no evidence they are adaptive; not a single TE insertion was functionally validated, so there is no proof any one of the insertions confers an adaptive value
- 316-318: again there is no time estimate when the TE emerged, yet they mention a narrow time-span

minor:

- 56 what do the authors exactly mean by plug and play units? TE mediated exon shuffling?
- 84 identifying adaptive TEs remain challenging due to the lack of large-scale studies. I would argue the main difficulty in identifying adaptive TEs is the lack of functional studies. it is easy to identify TE insertions and TE insertions differing in frequency among populations, but it is very hard to link this to adaptation.

Reviewer #3

(Remarks to the Author)

This paper presents a massive curated transposon dataset for *Zymoseptoria tritici*, one of the world's most important plant pathogens, causing septoria leaf blotch on wheat globally. This disease is manageable, but it demands huge fungicide inputs that incur large financial and environmental costs. Looking at genome sequences of ~2000 global isolates, the authors annotated over three million transposable elements. Having provided clear and detailed descriptions of their bioinformatic approaches, this serves as an excellent model for subsequent studies. Importantly, their work demonstrates correlations between SNP and TE variants and population differentiation, and loci that determine resistance to demethylase inhibiting compounds (mostly azoles), the most important class of fungicides, as well as succinate dehydrogenase inhibitors. Overall, I see this paper as a very nice extension of the excellent 2023 paper in Nature Communications by this group and others, that set up this deep dive into TEs (Feurtey et al. Nat. Comm. 14: 1059).

I thank the authors for providing an appropriately detailed and carefully written manuscript for review. I have a number of minor suggestions for improvement of the paper. I am not a bioinformaticist, and will focus on this from that of a biologist with indirect knowledge of this system and others like it.

Figures throughout: If at all possible, increase font sizes, and/or work on making the tiny fonts legible by using better contrast.

Figure 3: I think "associated with" is better than "leads to."

Figure 4: There is a lot of information here and the figure legend needs to be more precise and explicit. What are the dots? What are the orange numbers on the periphery of the diagram? I figured it out, but not stating it explicitly hinders the reader. Every type of symbol, shading, color, etc. needs to be described. Also, I suggest a more transparent shade for the red color in the inset, to enable reading the text – a more descriptive legend would also relieve the need for a separate legend for the inset, which for a moment I thought might be part of the figure. The word cloud is nice, but even blowing the figure up to 300% I struggle to read the orange text. Part of that is probably due to the colors that do not always contrast well. Please give this figure, and the important findings it demonstrates, a legend it deserves.

Figure 5: Use CE/BCE for both the timescale and the map – either that or BP throughout. The superscripts are very hard to read – perhaps these could be single letters, in parentheses and not superscripted.

Line 87: "considered universal" rather than "generalized"?

Last paragraph of introduction: Starts off in past tense, then switches to present tense.

Line 122: Choose "comprising" or "composed of"

Line 155: Delete "different"

Line 162: "...Australia and New Zealand do not exhibit this pattern, showing a significant..."

Lines 165-167: These time estimates seem quite reasonable to me, but there should be some external sources cited in support of them. The Feurtey et al. (2023) paper may be the correct source, or something preceding it?

Line 333: "Bar width..."

Line 378: "Co-occurrence..."

Line 389: principal

Line 425: I am with the authors on "derepression" as the likely best explanation for the patterns observed – but I think it's overstated in this sentence and subsequently. Perhaps it should be re-worded more as a supported hypothesis. Otherwise I really like these first few paragraphs in terms of laying out the significance of the study.

Line 453: "successful" – I'm not sure that's a good word to use for transpositions and nucleotide changes that, presumably, are often polymorphic in these populations?

Last paragraph: I can understand why the authors might not want to bring it up because it's somewhat of a can of worms, but these TEs are probably also introducing horizontally transferred genes as the fungus adapts to a new location. I do like the emphasis on transposition itself here and the points his paragraph is making with it.

Version 1:

Reviewer comments:

Reviewer #1

(Remarks to the Author)

I am pleased to recommend publication of this manuscript.

(Remarks on code availability)

I did not run everything in the github. I expect that would take a long time. I did review the codebase and I am able to determine that it would be straightforward to reproduce most of this analysis.

Reviewer #2

(Remarks to the Author)

The authors did a good job to address my comments.

While some things are still unclear, like the age of some insertions, and whether or not insertions are actually beneficial, I think on the overall the manuscript has improved.

Also I have to apologise for one comment, I realised too late that pangenome is a widely used term to refer to multiple genomes of a given species and that this does not necessarily imply pangenome graph.

I am aware that BayPass is widely used, but I still think there are dangers inferring selection just based on a single statistical approach. Especially when such approaches are statistically challenging than it is not feasible to intuitively identify potential sources of artefact. Also I'm not convinced that the explanation with covariance matrix ω is helpful to intuitively grasp the strength and limitations of this approach. Anyhow; I have no more comments that should be addressed.

(Remarks on code availability)

the code is available. for the future one thing that would be nice to integrate the figures into the github.md files. For example if figures are generated in Rmarkdown its easy to convert the entire document, including the resulting figures into a github.md

(output:

github_document)

Reviewer #3

(Remarks to the Author)

I read the responses to the reviews and the revised sections, and I am happy with the improvements.

(Remarks on code availability)

RESPONSE TO REVIEWERS

Reviewer #1 (Remarks to the Author):

Summary:

This study uses a large-scale dataset of genome sequences from thousands of fungal isolates to investigate the insertion dynamics of TEs during the species' colonization of globally widespread regions. Overall, the study is impressive in scope — thousands of samples and many TEs — with a clear, impactful headline detecting population-specific TE burst and potentially adaptive TE insertions. I have specific suggestions that I hope will improve the study and enhance the clarity of presentation. Overall I am supportive of publication.

Major:

Comment: *It is somewhat surprising that after manual curation, 105/331 (>30%) of Zymoseptoria TE models remain unclassified. Can the authors comment on this? Do the authors suspect that Zymoseptoria harbors TEs that mobilize through currently uncharacterized mechanisms?*

Response >>>> We thank the reviewer for raising this important point. Fungi have been severely understudied in terms of TE content and diversity, with just two species represented in the latest curated version of Dfam (v3.9, dfam.org). A lack of knowledge of TE diversity across fungi limits our ability to confidently classify all putative TE consensus sequences. Whilst several TE families could be confidently classified during manual curation using diagnostic features indicative of specific TE classifications (e.g. terminal inverted repeats, target site duplications, protein domains, long-terminal repeats, 3' poly-A tails, terminal sequence signatures), the unclassified consensus sequences lacked enough diagnostic evidence for a confident classification. Beyond these features, manual curation performed for establishing the TE classification for this work, also involved identifying homology to known TEs. However, given the lack of resources for Fungi and TE variability among even closely related species, we were unable to identify homology to well-curated elements. In these cases, many unclassified TEs were removed from the initial TE library due to matches with multicopy host genes, or due to their identification as segmental duplications. Those remaining have the potential to be true TEs, including potentially chimeric TEs, but lack currently-understood conserved diagnostic features or homology to curated elements (albeit from phylogenetically distant taxa). Some of these could be ancient inactive TE families that have degraded to the point where recognisable features are no longer present (e.g. through the process of repeat-induced point mutation), whilst others could be novel TE families or types that are currently not well understood. By including these putative TEs in the final TE library, we hope to be able to classify these as more knowledge and understanding of TE diversity across fungi becomes available (which is something we are also working on). We briefly mention these challenges when discussing TE classification.

Comment: *The authors should also provide the percent of the genome occupied by TEs in each class for each assembly used for TE model curation. This metric is useful because it is not affected by fragmentation issues in TE annotations, which*

can show biases across element types. Metrics such as “copy number” unfortunately suffer from this bias. Relatedly, occasional fragmented TE annotations may introduce redundant TE presence/absence calls. Can the authors comment on whether or not there is evidence of this in their data, and if so, how they address this challenge?

Response >>>> We have added a brief description of the range of total genome percentage occupied in these reference-quality genomes. In addition, we have provided a new Supplementary Figure S1 and Supplementary Table S1 with TE quantifications for these assemblies. Ordered by approximate colonisation timeline, we find a pattern of increasing TE content (as a percentage of genome size) (Figure S1; also provided below).

We agree that fragmentation is a major issue with TE annotation. In this case, Earl Grey was developed with specific steps to defragment TE annotations, which is implemented using RepeatCraft (see [10.1093/bioinformatics/bty745](https://doi.org/10.1093/bioinformatics/bty745)). In reviewer figure 1 (see below), we have compared TE copy number and TE genome coverage pre- and post-defragmentation. In line with the reviewer’s expectations, we show that TE copy number drops considerably following defragmentation, suggesting many TE annotations are fragmented, which can impact copy number estimates. Following defragmentation, we observed a smaller effect on TE genome coverage percentage, as there is a smaller change in the number of bases incorporated into defragmented annotations. For this study, we made use of defragmented annotations generated by Earl Grey using the manually curated library. We have added description of this to the materials and methods section.

Figure S1. Transposable element content among long-read reference-quality assemblies of the *Zymoseptoria tritici* global panel. Isolates are ordered based on rough colonisation history, with most ancient on the left and most recent on the right of the X axis. TE content is expressed as percentage of total genome size. Black line shows linear model of total genomic TE percentage by assembly (in order of colonisation), with the equation $y=0.187x + 18.274$, $R^2 = 0.41$, $p < 0.01$.

Reviewer Figure 1. TE quantification pre- and post-defragmentation. The top plot shows TE copy number for each reference-quality assembly annotated with Earl Grey, with quantifications before and after defragmentation is run with RepeatCraft, followed by resolution of overlapping repeats performed in Earl Grey post-processing (for post-defragmentation only). Bottom plots show TE copy number and TE coverage (as percentage of total assembly size) for the non-redundant assembly representing the global diversity of *Zymoseptoria tritici* (as used for McClintock2 calls in subsequent analyses).

Minor:

Comment: Transposable elements should be defined earlier in the introduction. They are currently mentioned first in the first paragraph and defined in the second paragraph

Response >>>> We have moved the definition of a TE to the first mention in the first paragraph.

Comment: Line 185: While I do not see a problem with separating DNA TEs and MITEs into separate groups throughout the MS, the authors should at least clearly

define the relationship between them. I am assuming that MITEs refer to nonautonomous “DNA TEs.”

Response >>>> This has been added at the first mention of MITEs.

Comment: *Some of the initial results go on a bit long describing the dataset and proximal results such as TE counts without getting to the main points of the paper. I suggest reducing some of the descriptive parts to supplemental tables.*

Response >>>> *We have reduced some of the initial results sections.*

Comment: *Line 392: Start sentence with “Three” rather than “3”*

Response >>>> We have implemented this change.

Comment: *I suggest reducing the amount of passive voice used throughout the methods to enhance readability.*

Response >>>> We have reduced the amount of passive voice throughout the materials and methods section.

###

Reviewer #2 (Remarks to the Author):

Baril et al., characterize TEs in >1900 publicly available genomes of Z. tritici, an important wheat pathogen. They identified >3million TE insertions. They claim that TE activity increased with the habitat expansion of the pathogen, and that this increased activity led to adaptive TE insertions. They have interesting data and this was clearly a massive effort, but I do not think that the data are strongly supporting any of these two claims. I do not see convincing evidence that TE activity increased during the last 500 years nor that any of the insertions have contributed to adaptation.

Response >>>> We appreciate the feedback and have provided point-by-point responses to each point below, which we believe has improved our manuscript. Briefly, we have performed additional analyses to determine whether the patterns we observed can be explained by either genetic drift and/or relaxed purifying selection. We have also ensured statistical tests are easier to find, and provided extra supplementary figures to support these analyses.

major

- where do I see that TE acitivity increased with global expansion of the pathogen? I guess in Figure 1. But upon comparing TE copy numbers in the ancestral populations (Iran and Israel) and the novel populations (Africa, America, Australia) the copy numbers seem very similar. Except perhaps in Australia where copy number is lower but this is contrary to their claim. The magnitude of the copy number differences seems in any case very small.

- 156 related to the above; they mention that copy numbers increased significantly but no *p*-value is provided (at least not in the main manuscript). Even if this is significant also the magnitude of the change should be provided; with very large data sets (1900 samples) even minor changes may be significant. As mentioned above the copy number increase seems very minor (except perhaps for LTR family_71)

Response >>>> Over the timescales (10-1000s of years) considered in this study, we do not expect to observe large TE copy number changes as a general pattern. However, we found some dramatic variations in individual TE family expansions, as we observe with the LTR family_71. Individual TE activation and proliferation has been demonstrated previously in an experiment conducted on the same species (10.1093/nar/gkad1214), where the Styx element has been independently reactivated in distinct *Z. tritici* populations, and evidence supports the activity of this element in a four-generation pedigree.

Despite the short time scales compared to between species TE comparisons, we identified significant differences in total TE copy number among populations, and provided Tukey post-hoc comparison statistics, including the difference in observed means, 95% confidence intervals, and adjusted *p*-values in Supplementary Table S1. Considering all significant pairwise comparisons, we find the mean difference in TE copy number between pairs of populations is 212 copies. This is e.g. much higher than previous observations in *Drosophila melanogaster*, where, for example, a difference of ~69.6 copies was observed between European and North American populations (10.1038/s41467-022-29518-8). When considering all pairwise, comparisons, both significant and non-significant, we observed a mean pairwise difference of 132 TE copies.

Considering significant pairwise comparisons, we find more recently established populations have higher TE copy numbers than more ancient populations (supplementary table S1; supplementary figure S2; reviewer figure 2). Specifically, the Africa & Southern Europe, Europe, USA & Canada, USA, and Argentina & Uruguay populations have significantly higher TE copy numbers than the Israel population at the centre of origin, consistent with an accumulation of TE content following the expansion from the centre of origin.

We have modified the main text to make the provision of these statistics clearer and also provided a new supplementary figure S2 (also provided here as reviewer figure 2) showing the significant pairwise comparisons with confidence intervals.

Reviewer Figure 2. Significant pairwise comparisons between populations with varying TE copy numbers, calculated using Tukey Honest Significant Differences, with an adjusted p -value significance cut-off $p = 0.05$. Pairwise comparisons are shown on the Y-axis, and difference between population means on the X-axis. Points indicate mean pairwise difference between populations, and bars indicate 95% confidence intervals. Negative changes indicate that the second population of each pair has a higher TE content, whilst positive changes indicate that the first population of each pair has a higher TE content.

Comment: - 161 they claim the data is consistent with increased activity, but where is evidence for this. an increase in TE copy number could also be due to founder effects, relaxed purifying selection etc, so there are several alternative hypothesis to consider for differences in TE copy numbers (which i do not see anyhow)

Comment: - 189 'Beyond changes in TE activity..' again I do not see evidence that the activity changed (could all be founder effects, drift...)

Response >>>> We agree that not any change in observed TE copy numbers necessarily reflects activation (*i.e.* activate transposition). Furthermore, TE activity is subject to the impacts of natural selection and genetic drift. Here, we provide additional analyses to further support the idea that TEs are indeed actively transposing beyond the published case of the *Styx* TE (10.1093/nar/gkad1214). The strongest evidence supporting ongoing TE activity is the identification of novel TE loci in derived populations. For a TE to be found at this locus, there must have been transposition activity resulting in a TE insertion at this site at some point in time. A confounding factor can be that a TE might have been already inserted at this specific locus but at such a low frequency that our population sequencing effort has missed it. Distinguishing between these two scenarios helps assess the evidence for TE activation.

To investigate the alternative hypothesis, we compared the TE locus frequencies within and between populations for the ten most abundant TE families, and *Styx*, for which previous evidence of activity exists (reviewer figures 3 & 4; supplementary figures S3 & S4). If the observed accumulation of TEs was due to founder effects (or relaxed purifying selection), the expectation would be to observe a shift in TE locus frequency towards higher frequency insertions. Contrary to this, for the arguably most active TE (LTR family_71), we observe an excess of very low-frequency insertions, suggesting that these TE loci are under purifying selection. We observe the same pattern for *Styx* (family_11), for which previous evidence of activity exists (10.1093/nar/gkad1214). Extending this analysis to the ten most abundant TE families across the global panel, we find the majority of TE insertions are fixed, but that more recent populations also show more than expected low frequency insertions, supportive of purifying selection acting on at least a subset of the TE loci.

Figure S3. TE locus occupancy for the putatively most active TE family (LTR family_71), the most abundant TE family in the centre of origin (LINE family_1507), and a TE family with experimental evidence for ongoing activity (*Styx* family_11). X

axis shows TE frequency for each locus. Y axis indicates the number of TE loci at a given frequency interval. Black lines show modelled frequency spectra based on global TE locus occupancy.

Supplementary Figure S4. TE locus occupancy for the ten most abundant TE families in the global panel of *Z. tritici*. X axis shows the percentage of isolates in a population containing a given TE locus. Y axis shows the locus count at a given population frequency. Each column shows TE frequency spectra for a given population, ordered from centre of origin to most recently derived population. Each

row shows TE frequencies for a given TE family. Red lines show expected TE frequency spectra based on global panel locus occupancy.

We assessed the likelihood that all TE insertions in derived populations could have been present in the centre of origin populations (*i.e.* no activation and pure genetic drift effects). We modelled expected TE locus frequency spectra based on TE locus frequencies observed in the global panel (red lines, reviewer figure 4; supplementary figure S4). If all possible TE insertions were found in the centre of origin, and not detected in our sample, we expect to find a higher low-frequency locus count than we did with our sampling. However, considering the top 10 most abundant TE families, we do not observe an excess of missed low-frequency insertions (reviewer figure 4, supplementary figure S4), except for the putatively most active LTR family_71. In the case of LTR family_71, we expect to find ~1,000 very low-frequency insertions in the centre of origin populations, which could explain some of the increased counts we find in more recently derived populations. However, we observe an excess of low-frequency insertions, inconsistent with undersampling in the centre of origin, in the Europe and USA populations, where ~8,000 and ~2,000 very low-frequency insertions are observed.

These findings are far more consistent with ongoing transposition activity in these populations generating new TE insertions rather than drift effects. The most parsimonious explanation for the observed patterns is therefore a balance of ongoing transposition and the subsequent effects of selection.

Genetic drift should also reduce the level of genetic diversity in a population. To test this, we compared the clustering of individuals in a population using TE loci and SNP loci (both filtered to a minor allele frequency > 0.05), using the first principal component computed from locus occupancy in each individual. If drift is the most dominant force shaping the TE landscape in these populations, we expect to observe similar clustering based on SNP and TE loci, as drift should act on both features similarly. Contrary to this expectation, we observe tighter clustering of individuals within a population using SNP loci, and a higher spread using TE loci (reviewer figure 5, supplementary figure S5). This is inconsistent with drift being the main force driving differences in TE landscapes among different populations, and is supportive of the observed landscapes resulting from ongoing transposition activity, and subsequent selection, leading to differential TE accumulation in distinct populations.

Reviewer Figure 5. Clustering of individuals within populations using either TE locus occupancy, or SNP occupancy, with loci filtered to minor allele frequency > 0.05. X axis indicates whether the principal component was computed using TE loci or SNP loci. Y axis shows normalised score for the first principal component. Each dot indicates an individual in a population, with connecting lines joining the same individual when using either TEs or SNPs. Focal population is shown in colour, as indicated in the key. In each panel, the non-focal populations are shown in grey.

We have added an additional paragraph to the results section acknowledging the potential roles of genetic drift (and selection) explaining our observations rather than TE activation. We also provide new supplementary figures with our new analyses to investigate these potential alternative explanations. We have also added mention of these processes to the discussion.

Comment: - 193-194 this is interesting; sub-populations from the USA sampled before 1990 contain <2000 TEs whereas subpopulations sampled in 2010 contain

more than 2000 TEs. so this could be evidence that TE activity increased recently. but with short read data it could also be due to problems with the data or the analysis, see this important paper <https://elifesciences.org/articles/28297> As a creative idea they used immobile elements (exons) to test their pipeline. It would be important if the authors also adapted this control. If their pipeline shows that the number of immobile elements (eg exons) is similar for the samples <1990 and 2010 but the numbers of TEs differs, then that would support increased activity in the USA between 1990 and 2010 (but than I do not get why activity just increased in the last decades, NA was colonized by *Z.tritici* around 1600 according to their fig1).

Response >>>> We appreciate the concern and would like to note that in the case of our study, we are characterising the TE landscape in a haploid organism grown from a single cell, and are not assessing a pool of somatic cells as in the cited work on *Drosophila*. Thus, we are focusing only on fixed insertions in a haploid, which can be much more robustly assessed. In *Z. tritici*, the dynamics of TE insertion and vertical inheritance are simpler, as every TE insertion has the potential to be passed on to the next generation, which is not the case with somatic mutations in organisms with a distinct germline. This also means that ongoing TE activity can have an immediate impact on the population, and that all successful insertions (i.e. non-lethal and not highly detrimental) could be retained and passed on.

Our study includes several benchmarking steps, which we customized to increase robustness. The peer-reviewed and benchmarked approaches provided in the McClintock2 suite of programs have been used extensively for characterisation of TE variation among populations. In addition to their own benchmarking, we performed our own robust benchmarking to determine the performance of each tool on our particular dataset (additional discussion is provided in additional file 1). We performed validation using simulation approaches, in which a false positive rate of 0 was found. In addition, further benchmarking using datasets where both long-reads and short-reads for the same isolates were available. The final decision on how to assess TEs was made based on tools that performed well across all benchmarking analyses.

The increase in TE content observed between isolates from the same Oregon site in 1990 and 2015 has been observed on smaller datasets in a previous study (10.7554/eLife.69249), in which it was confirmed that sequence coverage (a typical confounding factor) was not responsible for the observation.

Rather than a general increase in TE copy number between 1990 and 2015, we find certain families have expanded much more than others. This is consistent with TE activation as this should only affect a subset of TEs, which can be resurrected (Reviewer Table 1). Here, we see mean increases of 19.1 and 13.1 copies for the two most active LTR families over a period of 25 years. As TE activity is ongoing in *Z. tritici*, it is likely that some of these new insertions are the result of transposition activity. In line with the stress-induced transposition hypothesis, this activity could be triggered, for example, through environmental stress, such as sustained fungicide exposure. Intensive fungicide uses and rapid adaptation of *Z. tritici* could therefore explain the increase in TE copy number that we observe over this short time scale.

TE Family	TE Classification	Mean 1990	Mean 2015	Difference	P.value
ZymTri_2023_family_480	LTR	7	26.1	19.1	1.31e-55
ZymTri_2023_family_71	LTR/Copia	74.7	87.7	13.1	1.51e-10
ZymTri_2023_family_1473	Unclassified	13.7	25.4	11.7	1.45e-34
ZymTri_2023_family_495	RC/Helitron?	4.07	14.0	9.92	1.79e-54
ZymTri_2023_family_1222	LINE	13.6	22.0	8.42	3.86e-20
ZymTri_2023_family_99	LTR/Gypsy	4.57	12.0	7.45	8.64e-41
ZymTri_2023_family_346	DNA/MULE-MuDR	10.1	17.3	7.26	1.92e-32
ZymTri_2023_family_896	LTR/Gypsy	4.76	11.8	7.01	6.92e-33
ZymTri_2023_family_1347	MITE	44.5	51.4	6.87	2.84e-10
ZymTri_2023_family_1467	Unclassified	4.08	10.48	6.41	1.21e-34

Reviewer Table 1. The 10 TE families with the largest increase in copy number within isolates sampled in Oregon in 1990 and 2015. For each TE family, the difference among mean TE copy number is provided, as well as the P value from a student's T-test used to determine whether TE copy number has increased with significance determined as $p < 0.05$.

Comment: - *Fig2A it is quite unclear to me what 'affinity' means? as far as i guess it means overrepresentation or underrepresentation of TE families in the different z.tritici strains?*

Response >>>> We have modified the wording in figure 2 for clarity.

Comment: - *I think it should be made more clear early in the manuscript that the work is based on previously published data. It became clear to me only by looking at the material and methods.*

Response >>>> We further clarified that the genomes are publicly available in the introduction.

Comment: - *they claim they did a pangenome approach. which suggests to me, and perhaps to others a pangenome graph. But as far as i can tell the authors just did a de novo annotated TEs with EarlGrey in multiple related species and they call this pangenome approach, which is very misleading. This is overselling, especially given that people will likely associate this with pangenome graphs (e.g. using long-read assemblies) which indeed allows for a quite robust identification of TE polymorphism. But in contrast to this, their analysis is based on 1900 short-read data which allows for a far less robust identification of TEs insertions. Although I think they did a good job benchmarking the different tools for analysing the short read data.*

Response >>>> We regret the confusion regarding the approach. In this study, we generated a pangenome graph using REVEAL (<https://github.com/jasperlinthorst/reveal>), which is a graph-based multi-genome aligner. The resultant graph was made from the reference quality genome panel, where all genomes used were generated from long-read assemblies. We then extracted a non-redundant genome assembly representing the global diversity of *Z. tritici*, which is required for mapping of each short-read genome to call TE polymorphism.

The separate long-read assemblies were only used for generation and manual curation of the TE library for *Z. tritici*. It would be incomplete to say that we just performed a *de novo* TE annotation with Earl Grey. This was the first step prior to extensive manual curation to generate the final TE library (10.1186/s13104-023-06613-7), which included identification of genomic copies, characterisation of boundary sequences, protein-coding domains, similarity to known elements, and several rounds of refinement (see material and methods).

To clarify the use of the term "pangenome", we have altered the wording to make clear that we used a "non-redundant genome assembly representing the global diversity of *Z. tritici*".

Comment: - *258 the majority of the insertions occurred during the last 500 years. Where is the evidence for this? They did not calculate the age of any TE insertion. It is possible to compute the age of the insertions eg by LTR divergence or by the age of the haploblock around a TE, but they did not do this.*

Response >>>> The strongest evidence here comes from the likelihood that TE activity is ongoing due to the generation of novel insertions, as discussed in a previous response. The estimate of activity timing in this case is not linked to the age of specific TEs, but by identifying the populations in which the largest accumulations of elements are found. It is possible that some of these insertions have been lost in more ancient populations and retained in the more recently established ones.

In this study of ~2000 Illumina datasets (instead of PacBio/Nanopore), we are unfortunately unable to calculate the age of insertions using LTR divergence, as short-read mapping approaches define TE insertion based on reads overlapping the boundaries of host sequence and TE sequence, and thus we cannot characterise what segments of a TE are present when these boundaries are identified, only that a TE of a certain type is present.

Whilst it would be possible to assess the size of limited haploblocks, we feel that the uncertainty in reconstructing haplotype sequences near sites of active TEs would introduce significant uncertainty. Haploblock assessments would be far more realistic for more conserved (genic) regions.

We have modified the wording to account for the peak times to indicate peaks of accumulation.

Comment: - *Fig3 is interesting, but why are there so many TE loci just found in Europe?. Is this just because most samples are from Europe? What do we learn from this graph other than that most samples are from Europe and the 2nd most from the USA (though in a convoluted way)? I think it would be important to normalise the TE loci count by the number of samples from each region.*

Response >>>> The aim here was to quantify the number of TE loci that are specific to a given population, as support for ongoing TE activity generating distinct TE loci in different populations. In this case, an excess of TE loci found in a single population adds support for the hypothesis that ongoing TE activity is generating diversity in

distinct populations. We have added scaled TE locus counts to figure 3 to account for differences in sample size.

Comment: - *how were adaptive TEs identified? just with BayPass? but what is this tool doing, what kind of information is it utilizing to infer positive selection. Just mentioning BayPass is not sufficient for such a central message of this work. It needs to be very transparent how positively selected loci were identified, and what is the false positive rate of BayPass and to what kind of problems it is susceptible (e.g. population structure, sequencing errors). How was BayPass validated for this work? Is BayPass just using differences in the insertion frequencies among populations? Please consider alternative explanations such as founder effects and drift and how this could impact selection scans with BayPass.*

Response >>>> BayPass is a peer-reviewed tool widely used to identify loci likely under positive selection (10.1534/genetics.115.181453) in a geographically heterogeneous sample set. BayPass is designed to identify genetic markers subjected to selection and controls for population structure. The underlying models employed in BayPass explicitly account for covariance structure among the population allele frequencies resulting from the shared history of the populations, which is done using the covariance matrix Ω . Additionally, no assumptions are made regarding the underlying demography. The use of the covariance matrix makes the identification of variants subjected to selection less sensitive to the confounding effects of demography, and the calculation of the XtX statistic allows consideration of more complex population histories, including migration and ancestral admixture. Variants putatively under divergent selection are identified using the XtX statistic, which is explicitly corrected for the scaled covariance of population allele frequencies. In this case, significant outliers, based on the distribution of XtX values for each locus, are used to identify variants with strong signatures of adaptive differentiation.

A key strength of the BayPass methodology is linked to the identification of false positives. Hierarchically structured population histories have been shown to increase false positive rates, and BayPass overcomes this limitation by estimating the correlation structure of allele frequencies across the populations that originates from their shared histories. In benchmarking by the original authors, a false positive rate of close to 1% was identified when using the core model, which we employ in our study. In addition, the power (proportion of true positives among truly selected loci) was >99.9% for strongly associated variants (10.1534/genetics.115.181453).

We follow all recommended procedures for using BayPass, including multiple converging model runs, manually inspection and control for multiple testing, and applying an FDR-corrected significance threshold.

We have added more description to both the materials and methods, and to the appropriate results section describing the benefits of the BayPass approach.

Comment: - *280,286 etc adaptive TE loci: please write putatively adaptive, there is no evidence they are adaptive; not a single TE insertion was functionally validated, so there is no proof any one of the insertions confers an adaptive value*

Response >>>> We have modified the text as suggested.

Comment: -316-318: *again there is no time estimate when the TE emerged, yet they mention a narrow time-span*

Response >>>> These timespans reflect our observations of accumulation of specific TE families within specific populations, that have arisen at particular time frames given the documented colonization history of the pathogen. The establishment and detection of novel TE loci, that are not occupied in more ancient populations, combined with observed increases in TE family copy number, are supportive of a role for ongoing transposition activity, in combination with the effects of natural selection, leading to the observed patterns. In this case, the time estimates come from known timing of population establishment combined with the establishment of new TE insertion loci that are only found in a certain population to suggest this has arisen within that population and thus at a rough point in time (i.e. around the colonisation time). We clarify how we deduce information about time spans in the main text.

minor:

Comment: - 56 *what do the authors exactly mean by plug and play units? TE mediated exon shuffling?*

Response >>>> We have changed the wording to “protein-coding and regulatory units”.

Comment: - 84 *identifying adaptive TEs remain challenging due to the lack of large-scale studies. I would argue the main difficulty in identifying adaptive TEs is the lack of functional studies. it is easy to identify TE insertions and TE insertions differing in frequency among populations, but it is very hard to link this to adaptation.*

Response >>>> Yes, we agree. We also wanted to point out though that many smaller scale studies lacked the power to detect putatively adaptive TE insertions, which is a limitation a step prior to being able to functionally characterise candidates.

We obviously agree and have clarified that our search for loci with signatures of selection helps making inferences about selection pressure and adaptive evolution in the populations.

Reviewer #3 (Remarks to the Author):

Review of Nature Communications Manuscript 25-27639

*This paper presents a massive curated transposon dataset for *Zymoseptoria tritici*, one of the world's most important plant pathogens, causing septoria leaf blotch on wheat globally. This disease is manageable, but it demands huge fungicide inputs that incur large financial and environmental costs. Looking at genome sequences of ~2000 global isolates, the authors annotated over three million transposable elements. Having provided clear and detailed descriptions of their bioinformatic approaches, this serves as an excellent model for subsequent studies. Importantly,*

their work demonstrates correlations between SNP and TE variants and population differentiation, and loci that determine resistance to demethylase inhibiting compounds (mostly azoles), the most important class of fungicides, as well as succinate dehydrogenase inhibitors. Overall, I see this paper as a very nice extension of the excellent 2023 paper in Nature Communications by this group and others, that set up this deep dive into TEs (Feurtey et al. Nat. Comm. 14: 1059).

I thank the authors for providing an appropriately detailed and carefully written manuscript for review. I have a number of minor suggestions for improvement of the paper. I am not a bioinformaticist, and will focus on this from that of a biologist with indirect knowledge of this system and others like it.

Response >>>> We thank the reviewer for their positive comments and constructive feedback, which has helped to improve the manuscript and tighten our main messages. We have provided responses to each comment individually below, and sign-posted changes in the improved manuscript.

Comment: *Figures throughout: If at all possible, increase font sizes, and/or work on making the tiny fonts legible by using better contrast.*

Response >>>> We have made modifications to all figures to improve readability and clarity.

Comment: *Figure 3: I think “associated with” is better than “leads to.”*

Response >>>> We have updated the figure legend.

Comment: *Figure 4: There is a lot of information here and the figure legend needs to be more precise and explicit. What are the dots? What are the orange numbers on the periphery of the diagram? I figured it out, but not stating it explicitly hinders the reader. Every type of symbol, shading, color, etc. needs to be described. Also, I suggest a more transparent shade for the red color in the inset, to enable reading the text – a more descriptive legend would also relieve the need for a separate legend for the inset, which for a moment I thought might be part of the figure. The word cloud is nice, but even blowing the figure up to 300% I struggle to read the orange text. Part of that is probably due to the colors that do not always contrast well. Please give this figure, and the important findings it demonstrates, a legend it deserves.*

Response >>>> We agree and have modified the figure legend based on their suggestions. We have also modified the figure, changing the opacity of figure features to improve readability, and have modified text colours to improve readability.

Comment: *Figure 5: Use CE/BCE for both the timescale and the map – either that or BP throughout. The superscripts are very hard to read – perhaps these could be single letters, in parentheses and not superscripted.*

Response >>>> We have updated the figure, and also figure 1, to use BCE/CE. Superscripts have been replaced with single capital letters, and the key updated to reflect this change.

Comment: Line 87: “considered universal” rather than “generalized”?

Response >>>> We have implemented the suggested change.

Comment: Last paragraph of introduction: Starts off in past tense, then switches to present tense.

Response >>>> We have updated the last paragraph to use present tense.

Comment: Line 122: Choose “comprising” or “composed of”

Response >>>> We have incorporated this change.

Comment: Line 155: Delete “different”

Response >>>> We have incorporated this change.

Comment: Line 162: “...Australia and New Zealand do not exhibit this pattern, showing a significant...”

Response >>>> We have incorporated this change.

Comment: Lines 165-167: These time estimates seem quite reasonable to me, but there should be some external sources cited in support of them. The Feurtey et al. (2023) paper may be the correct source, or something preceding it?

Response >>>> We have added a source for these time estimates where these are mentioned in the text.

Comment: Line 333: “Bar width...”

Response >>>> We have incorporated this change.

Comment: Line 378: “Co-occurrence...”

Response >>>> We have incorporated this change

Comment: Line 389: principal

Response >>>> We have incorporated this change.

Comment: Line 425: I am with the authors on “derepression” as the likely best explanation for the patterns observed – but I think it’s overstated in this sentence and subsequently. Perhaps it should be re-worded more as a supported hypothesis. Otherwise I really like these first few paragraphs in terms of laying out the significance of the study.

Response >>>> We have reworded this section to present this as a hypothesis.

Comment: Line 453: “successful” – I’m not sure that’s a good word to use for transpositions and nucleotide changes that, presumably, are often polymorphic in these populations?

Response >>>> We have removed “successful” in this case.

Comment: Last paragraph: I can understand why the authors might not want to bring it up because it’s somewhat of a can of worms, but these TEs are probably also introducing horizontally transferred genes as the fungus adapts to a new location. I do like the emphasis on transposition itself here and the points his paragraph is making with it.

Response >>>> We agree that this is a relevant additional scenario. In the case of this manuscript, we aimed to focus on activation and activity because there is no evidence in this fungus for horizontal acquisition (yet).

REVIEWERS' COMMENTS

Reviewer #1 (Remarks to the Author):

I am pleased to recommend publication of this manuscript.

Reviewer #1 (Remarks on code availability):

I did not run everything in the github. I expect that would take a long time. I did review the codebase and I am able to determine that it would be straightforward to reproduce most of this analysis.

Reviewer #2 (Remarks to the Author):

The authors did a good job to address my comments.

While some things are still unclear, like the age of some insertions, and whether or not insertions are actually beneficial, I think on the overall the manuscript has improved.

Also I have to apologise for one comment, I realised too late that pangenome is a widely used term to refer to multiple genomes of a given species and that this does not necessarily imply pangenome graph.

I am aware that BayPass is widely used, but I still think there are dangers inferring selection just based on a single statistical approach. Especially when such approaches are statistically challenging than it is not feasible to intuitively identify potential sources of artefact. Also I'm not convinced that the explanation with covariance matrix ω is helpful to intuitively grasp the strength and limitations of this approach. Anyhow; I have no more comments that should be addressed.

Response: We agree and are grateful for the critical evaluation.

Reviewer #2 (Remarks on code availability):

the code is available. for the future one thing that would be nice to integrate the figures into the github.md files. For example if figures are generated in Rmarkdown its easy to convert the entire document, including the resulting figures into a github.md (output: github_document)

Response: Following the editor's request, we have not included figures.

Reviewer #3 (Remarks to the Author):

I read the responses to the reviews and the revised sections, and I am happy with the improvements.